# HDAC6 deacetylates ENKD1 to regulate mitotic spindle behavior and corneal epithelial homeostasis

Ting Song [1], Xueqing Han[1], Hanxiao Yin [2], Junkui Zhao[2], Mingming Ma[1], Xiaonuan Wen[1], Chunli Liu[1], Yiyang Yue[1], Huijie Zhao [1], Jun Zhou [1,2], Yang Yang [3]✉, Jie Ran [1]✉ & Min Liu [1,4]✉

## Abstract

**Corneal diseases can cause severe visual impairment and even blindness, which have been linked to the interruption of corneal epithelial homeostasis. However, the underlying molecular mechanisms are largely unknown. In this study, by comparing the transcriptomes of keratoconus, bacterial keratitis, viral keratitis, and healthy corneas, we found a steady upregulation of histone deacetylase 6 (HDAC6) in corneal diseases. Consistently, a significant increase in HDAC6 was observed in mouse corneas with bacterial keratitis. Overexpression of HDAC6 in mice results in a significant thickening of the corneal epithelium. Mechanistic studies reveal that HDAC6 overexpression disrupts mitotic spindle orientation and positioning in corneal epithelial cells. Our data further show that HDAC6 deacetylates enkurin domain-containing protein 1 (ENKD1) at lysine 98 and thereby impedes its interaction with γ-tubulin, restraining the centrosomal localization of ENKD1 and its proper function in regulating mitotic spindle behavior. These findings uncover a pivotal role for HDAC6-mediated deacetylation of ENKD1 in the control of corneal epithelial homeostasis, providing potential therapeutic targets for treating corneal diseases.**

**Keywords** Spindle Orientation; Spindle Positioning; Centrosome; Deacetylation; Corneal Epithelium
**Subject Categories** Cell Adhesion, Polarity & Cytoskeleton; Molecular Biology of Disease; Post-translational Modifications & Proteolysis

## Introduction

The cornea is the outermost layer of the eye, consisting of the corneal epithelium, stroma, endothelium, Bowman's membrane, and Descemet's membrane (Meek and Knupp, 2015; Ni et al, 2024). The cornea shields the eye from environmental germs, dust, and other harmful particles. Additionally, as a transparent structure, the cornea facilitates the passage of light into the eye and helps to focus light on the retina. This functional property is largely dependent on the ability of the corneal epithelium to undergo continuous renewal (Lin et al, 2023). The corneal epithelium serves as the eyeball's first barrier to the outside environment and is composed of outer 2–3 layers of flat squamous cells, inner 2–3 layers of wing cells, and a single layer of basal cells (Zhang et al, 2021). These cells are connected by tight junctions to protect the corneal interior from becoming infected by noxious environmental agents (Ban et al, 2003). The stratification and integrity of the corneal epithelium have been linked to the development of corneal diseases (Stephens et al, 2013). For example, the corneal epithelium thickens significantly in active herpetic stromal keratitis. However, the molecular mechanisms underlying corneal epithelial homeostasis remain largely unknown.

Accumulating evidence indicates that proper cell division is essential for the self-regulating property of the corneal epithelium (Castro-Muñozledo, 2013; Gupta and Chaudhuri, 2022). Mammalian cell division is orchestrated by the mitotic spindle, a structure composed of microtubules that emanate from centrosomes (Kozgunova et al, 2022). The centrosome is a membraneless organelle consisting of a pair of centrioles surrounded by the pericentriolar material (Blanco-Ameijeiras et al, 2022). In most cell types, the centrosome serves as the major microtubule organizing center to direct the formation of the bipolar spindle during mitosis (Fung et al, 2018; Theile et al, 2023; Wu et al, 2012). Mutation, abnormal expression, or mislocalization of centrosomal proteins have been shown to impair the spindle behavior, leading to defective cell division and tissue homeostasis (Decarreau et al, 2017; Hoffmann, 2021). Nevertheless, our current understanding of how the centrosome and spindle behavior are involved in corneal homeostasis remains very limited (Silverman et al, 2017; Waseem et al, 2020).

Histone deacetylase 6 (HDAC6), a centrosome/microtubule-associated deacetylase that resides predominantly in the cytoplasm and participates in diverse biological processes such as cell migration, cilium disassembly, and axon growth (Kalinski et al, 2019; Lafarga et al, 2012; Ran et al, 2015), has recently been implicated in meiotic spindle assembly in mouse oocytes (Ling et al, 2018; Zhou et al, 2017). However, it remains unclear whether HDAC6 plays a role in controlling the mitotic spindle behavior to maintain corneal epithelial homeostasis. In this study, we

[1]Center for Cell Structure and Function, Shandong Provincial Key Laboratory of Animal Resistance Biology, College of Life Sciences, Shandong Normal University, 250014 Jinan, China. [2]Department of Genetics and Cell Biology, State Key Laboratory of Medicinal Chemical Biology, College of Life Sciences, Nankai University, 300071 Tianjin, China. [3]Translational Medicine Center, The First Affiliated Hospital of Zhengzhou University, 450052 Zhengzhou, China. [4]Laboratory of Tissue Homeostasis, Haihe Laboratory of Cell Ecosystem, 300462 Tianjin, China. ✉E-mail: fccyangyang430@zzu.edu.cn; jran@sdnu.edu.cn; minliu@nankai.edu.cn

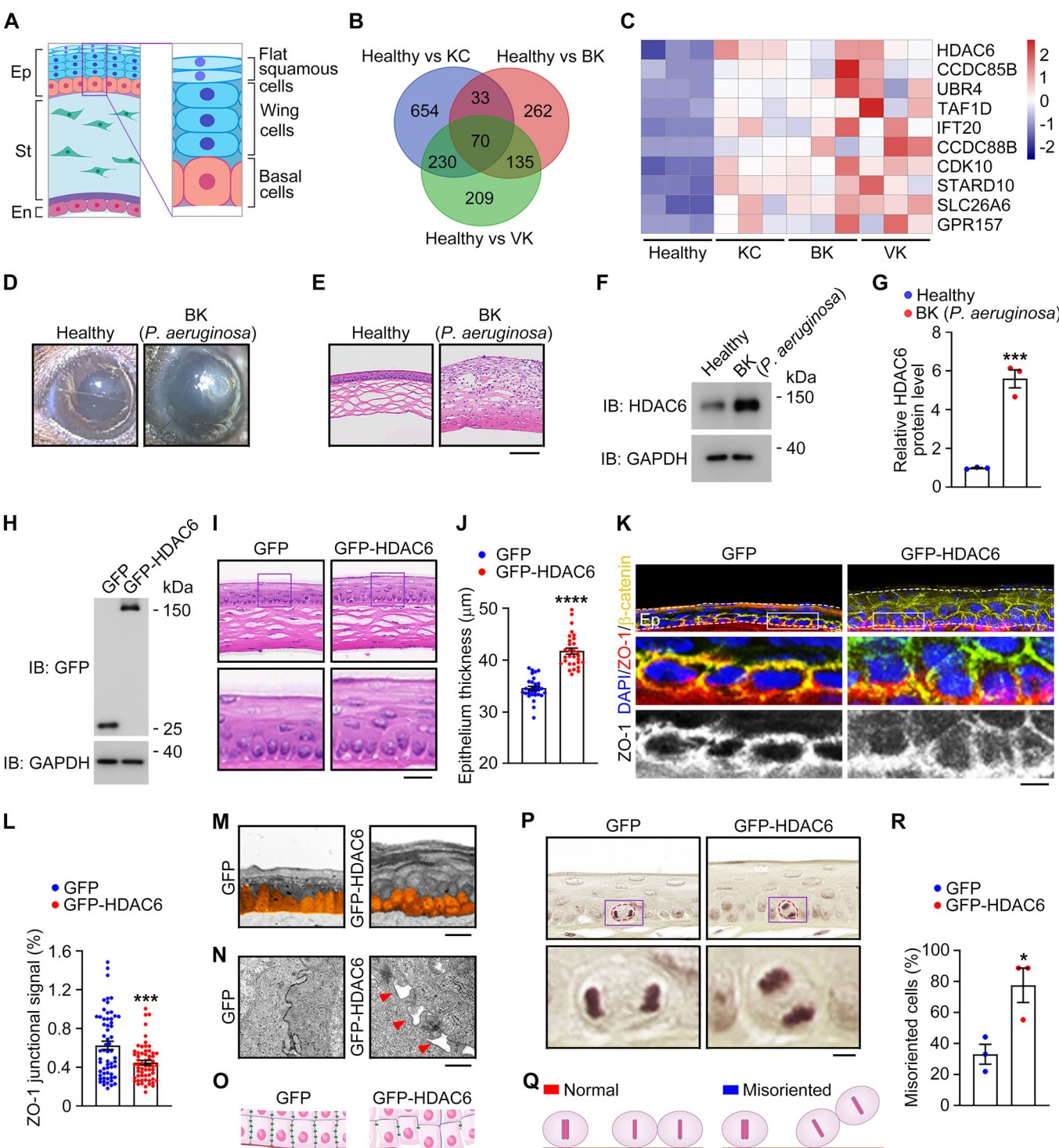

demonstrate that the level of HDAC6 expression is abnormally elevated in multiple corneal diseases. Overexpression of HDAC6 in mouse corneas results in a significant thickening of the corneal epithelium. Mechanistically, our data reveal that HDAC6 deacetylates enkurin domain-containing protein 1 (ENKD1) to control mitotic spindle orientation and positioning in corneal epithelial cells.

## Results

### HDAC6 expression is upregulated in corneal diseases

The corneal epithelium functions as a barrier to protect the eye from external hazards and provides a smooth surface for maximizing optical quality (Fig. 1A). Defects in the corneal

◄ **Figure 1.** HDAC6 overexpression disrupts corneal epithelial homeostasis.

(A) Schematic illustration of the corneal epithelium (Ep), stroma (St), and endothelium (En). (B) Venn diagram showing the number of genes differentially regulated in the pairwise comparison of healthy, keratoconus (KC), bacterial keratitis (BK), and viral keratitis (VK) corneas. The mRNA microarray dataset GSE241715 from the GEO database was used in the comparison. (C) Heatmap showing genes differentially regulated in the centrosome ($P < 0.05$, log2foldchange $> +1$). (D) Representative images of corneas from healthy and BK (*P. aeruginosa*) mice. (E) H&E staining images of corneas from healthy and BK (*P. aeruginosa*) mice. Scale bar, 70 µm. (F, G) Immunoblotting (F) and quantification (G, $n = 3$ independent experiments) showing the level of HDAC6 in corneas from healthy and BK (*P. aeruginosa*) mice. The intensity of the HDAC6 band was normalized to that of the GAPDH band. $P = 0.0006$. (H) Immunoblot analysis of GFP and GAPDH in corneas from control GFP or GFP-HDAC6 adenovirus-injected mice. (I, J) H&E staining images (I) and quantification of the thickness of the epithelium (J, $n = 30$ images from three independent experiments) from control or HDAC6 adenovirus-injected mice. Scale bar, 15 µm. $P < 0.0001$. (K, L) Immunofluorescence images (K) and quantification of the ZO-1 junctional signals (L, $n = 60$ cells from three independent experiments) of the cornea from control or HDAC6 adenovirus-injected mice, stained with antibodies against β-catenin and ZO-1 and DAPI. Scale bar, 30 µm. $P = 0.0004$. (M, N) Transmission electron microscopy images of the longitudinal sections of corneal epithelial cells (M) and cell junctions at the base of the corneal epithelium (N) in control and HDAC6 adenovirus-injected mice. Scale bars, 15 µm (M) and 500 nm (N). (O) Schematic illustration of basal cell junctions in the corneal epithelium from control or HDAC6 adenovirus-injected mice. (P) H&E staining images of the corneal epithelium from control or HDAC6 adenovirus-injected mice. Scale bar, 4 µm. (Q, R) Schematic illustration (Q) and quantification (R, $n = 15$ mitotic cells from three independent experiments) of normal and misoriented cell division in the corneal epithelium from control or HDAC6 adenovirus-injected mice. $P = 0.0257$. Data are presented as mean ± SEM. Unpaired two-tailed Student's *t* test was performed. *$P < 0.05$; ***$P < 0.001$; ****$P < 0.0001$. Source data are available online for this figure.

epithelium can cause a variety of corneal diseases, such as keratitis, keratoconus, and corneal dystrophy. To identify potential factors involved in the development of corneal diseases, the mRNA microarray dataset GSE241715, which characterizes the molecular profiles of healthy, keratitis, and keratoconus corneas, was obtained from the Gene Expression Omnibus (GEO) database (Lapp et al, 2023; Data ref: Lapp et al, 2023). Through pairwise comparison, we initially identified 654 [healthy *vs.* keratoconus (KC) individuals], 262 [healthy *vs.* bacterial keratitis (BK) individuals], and 209 [healthy *vs.* viral keratitis (VK) individuals] significantly upregulated genes (Figs. 1B and EV1A), respectively. Venn diagram analysis revealed 70 genes present in the three comparison datasets (Fig. 1B). Subsequently, we performed gene ontology (GO) enrichment analysis to explore the pathways associated with these 70 overlapping genes. The result of GO enrichment analysis demonstrated that the functions of these gene products were primarily concentrated in the categories of "centrosome", "negative regulation of cell growth", and "actin-based cell projection" (Fig. EV1B). Interestingly, heatmap visualization revealed that HDAC6 is highly upregulated in corneal diseases compared to healthy corneas (Fig. 1C).

To verify the above findings, we utilized the PAO1 strain of *Pseudomonas aeruginosa* (*P. aeruginosa*) to establish a mouse model of BK (Minns et al, 2023). One day following *P. aeruginosa* infection, the corneas of the BK mice exhibited significant opacification (Fig. 1D). We then examined the whole corneal histology through microscopic analysis of hematoxylin and eosin (H&E)-stained sections. We observed typical pathological changes associated with BK, including corneal thickening, infiltration of epithelial cells into the stroma, and disorganization of the corneal structure (Fig. 1E), indicating that *P. aeruginosa* infection substantially disrupts corneal homeostasis. Consistent with this finding, reverse transcription quantitative PCR (RT-qPCR) analysis demonstrated that the mRNA levels of various pro-inflammatory cytokines, such as *Il-6*, *Il-10*, and *Il-1β* were significantly elevated in the BK mice compared to controls (Fig. EV1C–E). Subsequently, we examined the HDAC6 protein level and found that it was significantly higher in the BK mice compared to controls, confirming the GEO dataset analysis results (Fig. 1F,G). Collectively, these results demonstrate HDAC6 overexpression in corneal diseases, particularly in BK, and indicate a potential role for this protein in regulating corneal epithelial homeostasis.

## HDAC6 regulates corneal epithelial homeostasis by modulating the orientation of cell division

To decipher whether the upregulation of HDAC6 affects corneal homeostasis, mice were intracamerally injected with adenoviruses encoding GFP-HDAC6. Corneal GFP-HDAC6 expression escalated dramatically 10 days post-injection with the adenoviruses (Figs. 1H and EV1F), and the level of exogenous HDAC6 was substantially higher than endogenous HDAC6 in the cornea (Fig. EV1G,H). We observed a significant thickening of the corneal epithelium upon HDAC6 overexpression (Fig. EV1F). To ascertain the alteration in corneal epithelial structure, H&E-stained corneal sections were examined through microscopic analysis. We found a considerable increase in corneal epithelial thickness in mice injected with HDAC6 adenoviruses compared to those receiving control adenoviruses (Fig. 1I,J). However, no discernible changes were observed in the corneal stromal and endothelial tissues upon HDAC6 overexpression (Figs. 1I and EV1I,J), highlighting the specific disruption of the corneal epithelial architecture.

We next investigated the effects of HDAC6 deficiency on corneal homeostasis using *Hdac6* knockout mice, in which exons 10–13 of the *Hdac6* gene were deleted by homologous recombination (Fig. EV1K). The absence of HDAC6 in the cornea derived from the knockout mice was validated by immunoblotting (Fig. EV1L). Subsequently, we examined the whole corneal histology by microscopic analysis of H&E-stained sections. The cornea of *Hdac6* knockout mice did not exhibit apparent changes compared to wild-type mice (Fig. EV1M–P), presumably due to the redundancy of functions shared with other HDACs (Minns et al, 2023).

To elucidate the role of HDAC6 in the corneal epithelium, we stained corneal sections with antibodies against β-catenin, a marker of epithelial cells, and zonula occludens-1 (ZO-1), a component of the tight junction complex. Immunofluorescence microscopy demonstrated severe abnormalities in corneal epithelial morphology (Fig. 1K). Notably, HDAC6 overexpression severely disrupted the tight junctions between corneal epithelial cells (Fig. 1K,L), indicating a potential impairment of the barrier function. Strikingly, we detected significant irregularities in the organization of corneal epithelial cells in HDAC6 adenovirus-injected mice. In the control cornea, epithelial cells were neatly organized horizontally along the upper edge. In contrast, upon HDAC6 overexpression, corneal epithelial cells became disorganized (Fig. 1K), suggesting a

potential disruption of corneal epithelial homeostasis. To corroborate these findings, we examined the longitudinal sections of the corneal epithelium with transmission electron microscopy. Both the organization and tight junctions of the corneal epithelium in HDAC6 adenovirus-injected mice displayed varying degrees of abnormalities (Fig. 1M–O). Therefore, intracameral injection of HDAC6 adenoviruses impairs corneal epithelial homeostasis.

To study how HDAC6 is involved in corneal epithelial homeostasis, we investigated whether the cell cycle progression of epithelial cells was affected upon HDAC6 overexpression. Human corneal epithelial (HCE-2) cells were treated with adenoviruses encoding GFP-HDAC6. Flow cytometry showed that the distribution of cells in G1, S, and G2/M phases remained unchanged upon HDAC6 overexpression (Fig. EV2A,B). Additionally, we examined the expression level of cyclin-dependent kinase 10 (CDK10), a cell cycle regulatory protein that was identified by our GEO dataset analysis to be associated with corneal diseases. However, RT-qPCR and immunoblotting showed no alterations in the mRNA or protein levels of CDK10 upon HDAC6 overexpression (Fig. EV2C–E). These findings indicate that HDAC6 overexpression does not influence the cell cycle progression of corneal epithelial cells. Subsequently, we assessed the number of EdU-positive cells in HDAC6-overexpressing HCE-2 cells and found that it did not differ from that of controls (Fig. EV2F,G), suggesting that HDAC6 overexpression does not affect the proliferation of corneal epithelial cells.

The orientation of cell division plays a crucial role in the proper organization and shaping of tissues (Lechler and Mapelli, 2021). We thus evaluated the impact of HDAC6 on cell division orientation through microscopic analysis of H&E-stained corneal epithelial sections. In control corneas, epithelial cells exhibited symmetric division, and their orientation was parallel to the basement membrane. In contrast, in corneas treated with HDAC6 adenoviruses, the orientation of epithelial cell division deviated from the basement membrane (Fig. 1P–R). This finding indicates that overexpression of HDAC6 leads to the misorientation of the division of corneal epithelial cells, resulting in the thickening of the corneal epithelium. We also analyzed the architecture of the corneal epithelium in *Hdac6* knockout mice by immunofluorescence staining. We found that ablation of HDAC6 does not compromise the integrity of tight junctions between corneal epithelial cells, nor does it alter the morphology or polarity of these cells (Fig. EV2H,I). These findings are consistent with our H&E-staining results showing that depletion of HDAC6 does not significantly affect the corneal architecture (Fig. EV1M–P).

## The deacetylase activity of HDAC6 is important for its action on spindle orientation and positioning

Proper orientation of cell division is largely dependent on the accurate orientation of the mitotic spindle. To determine whether overexpression of HDAC6 disrupts spindle orientation, we measured the angle between the spindle axis and the substratum in HeLa cells transfected with HA-HDAC6 (Fig. 2A,B). Time-lapse microscopy revealed that the spindle angle in HDAC6-overexpressing HeLa cells underwent more dramatic changes before anaphase onset, compared to cells transfected with the HA vector (Fig. 2C,D). Additionally, we examined HCE-2 cells transfected with GFP-HDAC6 (Fig. EV3A,B). We found that HDAC6 overexpression in HCE-2 cells significantly increased the average spindle angle (from less than 10° to over 30°) and

broadened the spindle angle distribution, although the overall morphology and length of the spindle were not obviously affected (Figs. 2E,F and EV3C,D). These results suggest that enhanced expression of HDAC6 stimulates spindle misorientation in corneal epithelial cells. We also investigated the potential effect of HDAC6 on spindle positioning, by measuring the distance between the spindle center and the cell center (referred to as spindle displacement distance, Fig. 2G). We found that enhanced expression of HDAC6 did not obviously affect cell diameter, but resulted in a significant increase in spindle displacement distance (Figs. 2H,I and EV3E), indicating spindle positioning defects. Collectively, the above results reveal an important function for HDAC6 in the control of spindle orientation and positioning in corneal epithelial cells.

As a member of the HDAC family, HDAC6 has been shown to regulate biological processes by deacetylating various substrate proteins. Thus, we analyzed whether the function of HDAC6 in spindle orientation and positioning is dependent on its deacetylase activity. HCE-2 cells were treated with tubacin, a small-molecule compound specifically inhibiting the deacetylase activity of HDAC6 (Fig. EV3F). We found that tubacin had no obvious effect on the spindle angle or spindle displacement distance in HCE-2 cells (Fig. EV3G–L). Consistent with this observation, the spindle behavior in HDAC6-depleted HCE-2 cells was not significantly affected (Appendix Fig. S1A–I). However, when cells with enhanced expression of HDAC6 were subjected to tubacin treatment, the deleterious effects on spindle orientation and positioning were significantly attenuated (Figs. 2J–M and EV3M–Q). These findings thus suggest that the deacetylase activity of HDAC6 is important for its role in spindle behavior.

To corroborate the above findings, HDAC6-depleted HCE-2 cells were transfected with various mutants of HDAC6, including H215A (histidine 215 mutated to alanine in the first deacetylase domain), H610A (histidine 610 mutated to alanine in the second deacetylase domain), and H215/610A (mutations of both histidine 215 and histidine 610 to alanines). We found that H610A and H215/610A remarkably abolished the deacetylase activity of HDAC6 in HCE-2 cells, whereas the H215A mutant had a relatively minor effect (Appendix Fig. S2A). In agreement with this result, defects in spindle orientation and positioning were obviously detected in cells transfected with wild-type HDAC6 or the H215A mutant, but not in cells transfected with the H610A or H215/H610A mutant (Fig. 2N–Q; Appendix Fig. S2B,C). Collectively, these results suggest that the effect of HDAC6 on spindle orientation and positioning in corneal epithelial cells is dependent on its deacetylase activity.

HDAC6-mediated deacetylation of α-tubulin plays an important role in multiple microtubule-dependent processes, such as axon growth and cilium disassembly (Ran et al, 2015; Saunders et al, 2022). To investigate whether α-tubulin deacetylation is involved in HDAC6-mediated spindle orientation and positioning, HCE-2 cells were transfected with HA-HDAC6, together with GFP-α-tubulin wild-type or its mutants, including K40Q (mutation of lysine 40 to glutamine to mimic acetylation) and K40R (mutation of lysine 40 to arginine to disrupt acetylation). However, no significant change in the spindle behavior was detected in HCE-2 cells transfected with GFP-α-tubulin wild-type or its mutants compared to those transfected with GFP (Appendix Fig. S2D–I). These results indicate that the activity of HDAC6 in regulating the spindle behavior is not mediated by its deacetylation of α-tubulin.

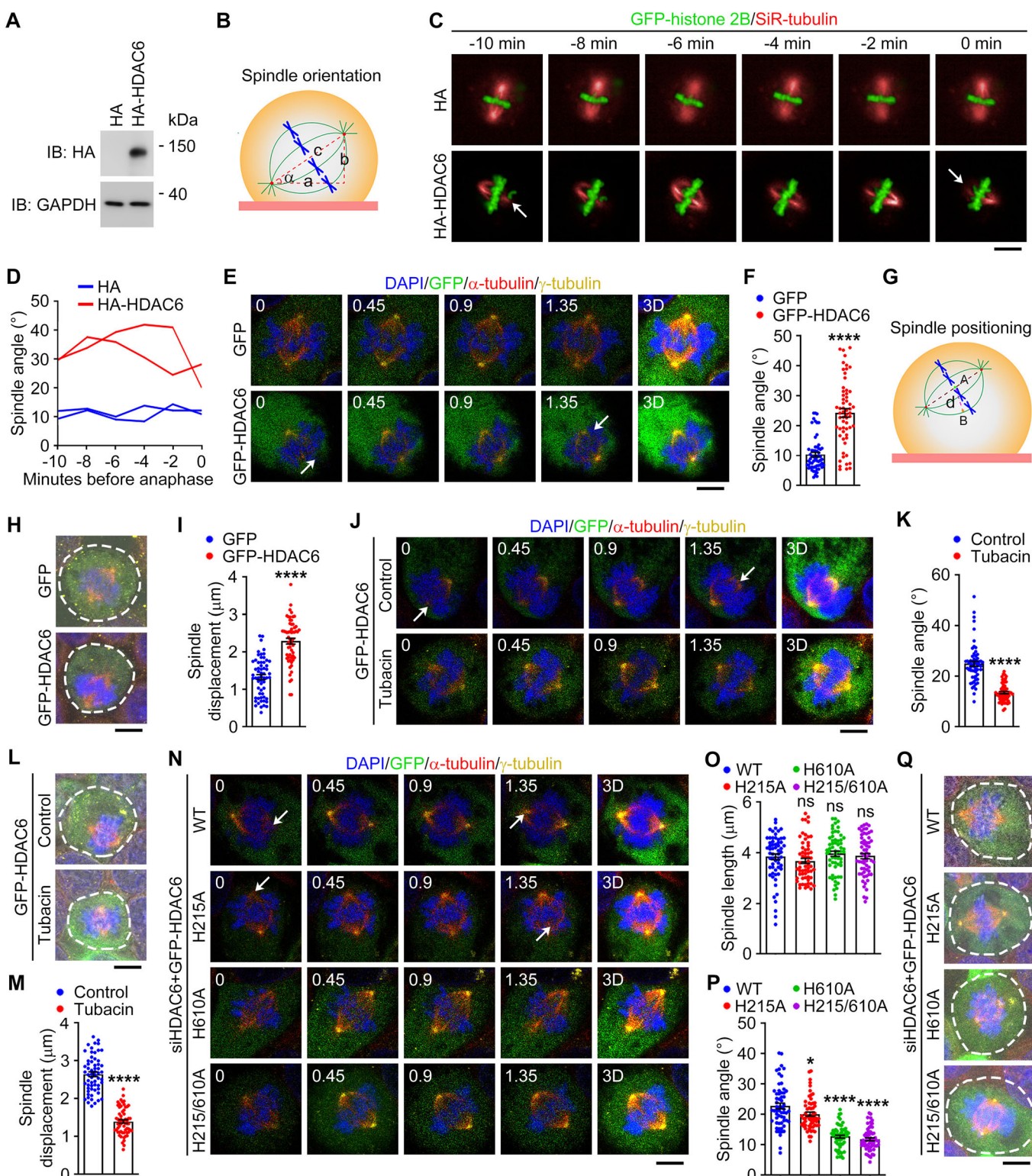

## HDAC6 interacts with ENKD1 within the centrosome

To further characterize the activity of HDAC6 in regulating spindle orientation and positioning, we examined the protein level and subcellular localization of HDAC6 in corneal epithelial cells throughout the cell cycle. We found that the HDAC6 protein level was almost constant during the cell cycle (Fig. 3A). Consistent with our previous results and those of others (Ran et al, 2015; Sánchez de Diego et al, 2014; Wang et al, 2013), we found a prominent localization of HDAC6 at the centrosome in interphase

Figure 2. The deacetylase activity of HDAC6 is required for spindle orientation and positioning in corneal epithelial cells.

(A) Immunoblot analysis of HA and GAPDH in HeLa cells transfected with HA or HA-HDAC6. (B) Schematic illustration for the measurement of spindle angle (α). (C, D) Time-lapse images (C, zx projection) and spindle angles (D) in mitotic HeLa cells transfected with GFP-histone 2B and HA or HA-HDAC6. Microtubules were stained with SiR-tubulin (red). Anaphase onset was set at 0 min. The white arrow refers to the spindle pole that does not appear on the corresponding focal plane (C). Scale bar, 6 μm. (E, F) Immunofluorescence images (E) and quantification of spindle angle (F, $n = 60$ images from three independent experiments) of metaphase HCE-2 cells transfected with GFP-HDAC6 or GFP, and stained with antibodies against α-tubulin and γ-tubulin and DAPI. The white arrow refers to the spindle pole that does not appear on the corresponding focal plane (E). Scale bar, 4 μm. $P < 0.0001$. (G) Schematic illustration for the measurement of spindle displacement distance (d). (H, I) Immunofluorescence/bright-field images (H) and quantification of spindle displacement distance (I, $n = 60$ images from three independent experiments) of metaphase HCE-2 cells transfected with GFP-HDAC6 or GFP, and stained with antibodies against α-tubulin and γ-tubulin and DAPI. The white circle indicates the cell boundary (H). Scale bar, 5 μm. $P < 0.0001$. (J, K) Immunofluorescence images (J) and quantification of spindle angle (K, $n = 60$ images from three independent experiments) from metaphase HCE-2 cells transfected with GFP-HDAC6 and treated with tubacin (10 μM), and stained with antibodies against α-tubulin and γ-tubulin and DAPI. The white arrow refers to the spindle pole that does not appear on the corresponding focal plane (J). Scale bar, 4 μm. $P < 0.0001$. (L, M) Immunofluorescence/bright-field images (L) and quantification of spindle displacement distance (M, $n = 60$ images from three independent experiments) from metaphase HCE-2 cells transfected with GFP-HDAC6 and treated with tubacin (10 μM), and stained with antibodies against α-tubulin and γ-tubulin and DAPI. The white circle indicates the cell boundary (L). Scale bar, 5 μm. $P < 0.0001$. (N–P) Immunofluorescence images (N) and quantifications of spindle length (O, $n = 60$ images from three independent experiments) and spindle angle (P, $n = 60$ images from three independent experiments) from metaphase HCE-2 cells transfected with HDAC6 siRNAs and GFP-HDAC6 wild-type, H215A, H610A, or H215/610A, and stained with antibodies against α-tubulin and γ-tubulin and DAPI. The white arrow refers to the spindle pole that does not appear on the corresponding focal plane (N). Scale bar, 4 μm. Spindle length: H215A, $P = 0.55$; H610A, $P = 0.7609$; H215/610A, $P = 0.9961$. Spindle angle: H215A, $P = 0.0154$; H610A, $P < 0.0001$; H215/610A, $P < 0.0001$. (Q) Immunofluorescence/bright-field images of metaphase HCE-2 cells transfected with HDAC6 siRNAs and GFP-HDAC6 wild-type, H215A, H610A, or H215/610A, and stained with antibodies against α-tubulin and γ-tubulin and DAPI. The white circle indicates the cell boundary. Scale bar, 5 μm. Data are presented as mean ± SEM. Non-parametric one-way ANOVA with post hoc analysis was performed for (O, P). Unpaired two-tailed Student's $t$ test was performed for (F, I, K, M). *$P < 0.05$; ****$P < 0.0001$; ns not significant. Source data are available online for this figure.

HCE-2 cells, as revealed by immunostaining of HDAC6 together with the centrosomal marker γ-tubulin (Fig. 3B,C). However, during metaphase, HDAC6 was predominantly dispersed throughout the cytoplasm, and its localization at the centrosome/spindle poles was reduced (Fig. 3B,C). Furthermore, we assessed the localization of exogenous HDAC6 at the centrosome. We found that the localization of GFP-HDAC6 at the centrosome was also weakened during metaphase, but this change was relatively modest compared to that observed with endogenous HDAC6 (Appendix Fig. S3A,B). These results indicate that the reduced localization of HDAC6 at the centrosome is crucial for maintaining proper spindle behavior, and that overexpression of HDAC6 may interfere with this dynamic localization and lead to abnormal spindle behavior. Based on these observations and our demonstration of the importance of HDAC6 deacetylase activity in the control of spindle orientation and positioning, we speculated that HDAC6 might play a role in these processes through the modulation of downstream centrosomal proteins.

To test this possibility, we sought to identify centrosomal proteins that interact with HDAC6 in corneal epithelial cells. Mass spectrometric analysis of proteins present in the HDAC6 immunoprecipitate identified ENKD1, a centrosome-associated protein (Song et al, 2022a; Tiryaki et al, 2022; Zhong et al, 2022), as an HDAC6-binding partner (Fig. 3D,E). Additional immunoprecipitation experiments confirmed that ENKD1 interacted with HDAC6 in corneal lysates and HCE-2 cells (Fig. 3F,G). In addition, immunoprecipitation revealed that GFP-ENKD1 interacted with HA-HDAC6 in HEK293T human embryonic kidney epithelial cells (Fig. 3H). Using a series of truncated HDAC6 and ENKD1 constructs, we further found that both the first and the second deacetylase domains of HDAC6 were required for its interaction with ENKD1 and that the centrosomal domain of ENKD1 was important for its interaction with HDAC6 (Fig. 3I,J). In addition, GST pull-down analysis with purified GST-HDAC6 and His-ENKD1 showed a direct interaction of these two proteins in vitro (Fig. 3K).

We then sought to examine the localization of ENKD1 in corneal epithelial cells. By immunofluorescence microscopy, we found that ENKD1 was localized to the centrosome in interphase HCE-2 cells.

Intriguingly, we observed a remarkable increase in ENKD1 localization at the centrosome/spindle pole in mitotic HCE-2 cells (Fig. 3L,M). Given the critical role of ENKD1 in mediating centrosome-associated processes and its direct interaction with HDAC6, our results indicate that ENKD1 localization at the centrosome/spindle pole in corneal epithelial cells might be closely linked to the modulation of spindle orientation and positioning by HDAC6.

## HDAC6-mediated deacetylation regulates the centrosomal localization of ENKD1

To determine whether ENKD1 is a substrate of HDAC6, we transfected Flag-ENKD1 and GFP-HDAC6 in HEK293T cells. We found a significant reduction in ENKD1 acetylation in cells transfected with HDAC6 wild-type and the H215A mutant, and a slight reduction in cells transfected with the H610A mutant; however, the H215/H610A mutant did not obviously affect ENKD1 acetylation (Figs. 4A and EV4A). Conversely, inhibition of HDAC6 activity with tubacin or knockdown of HDAC6 expression with two different siRNAs could efficiently increase the acetylation of ENKD1 (Figs. 4B,C and EV4B,C). Additionally, corneas from mice injected with HDAC6 adenoviruses exhibited a significantly lower level of ENKD1 acetylation compared to the control group (Figs. 4D and EV4D). Taken together, these results indicate that HDAC6 promotes the deacetylation of ENKD1.

Using the CSS-Palm software, we found five lysine residues within ENKD1 as potential acetylation sites (Fig. 4E). As ENKD1 is an evolutionarily conserved protein, we speculated that the acetylated lysines might be conserved during evolution. Alignment of human and mouse ENKD1 sequences revealed that the five predicted lysine residues are indeed highly conserved (Fig. 4F). Subsequently, we engineered acetylation-deficient mutants of ENKD1 by substituting the lysine residues with arginines (K95R, K98R, K131R, K193R, and K343R). Immunoprecipitation experiments revealed that only the K98R mutations impeded the deacetylation of ENKD1 by HDAC6 (Fig. 4G,H), indicating that K98 is the essential site for HDAC6-mediated deacetylation. We

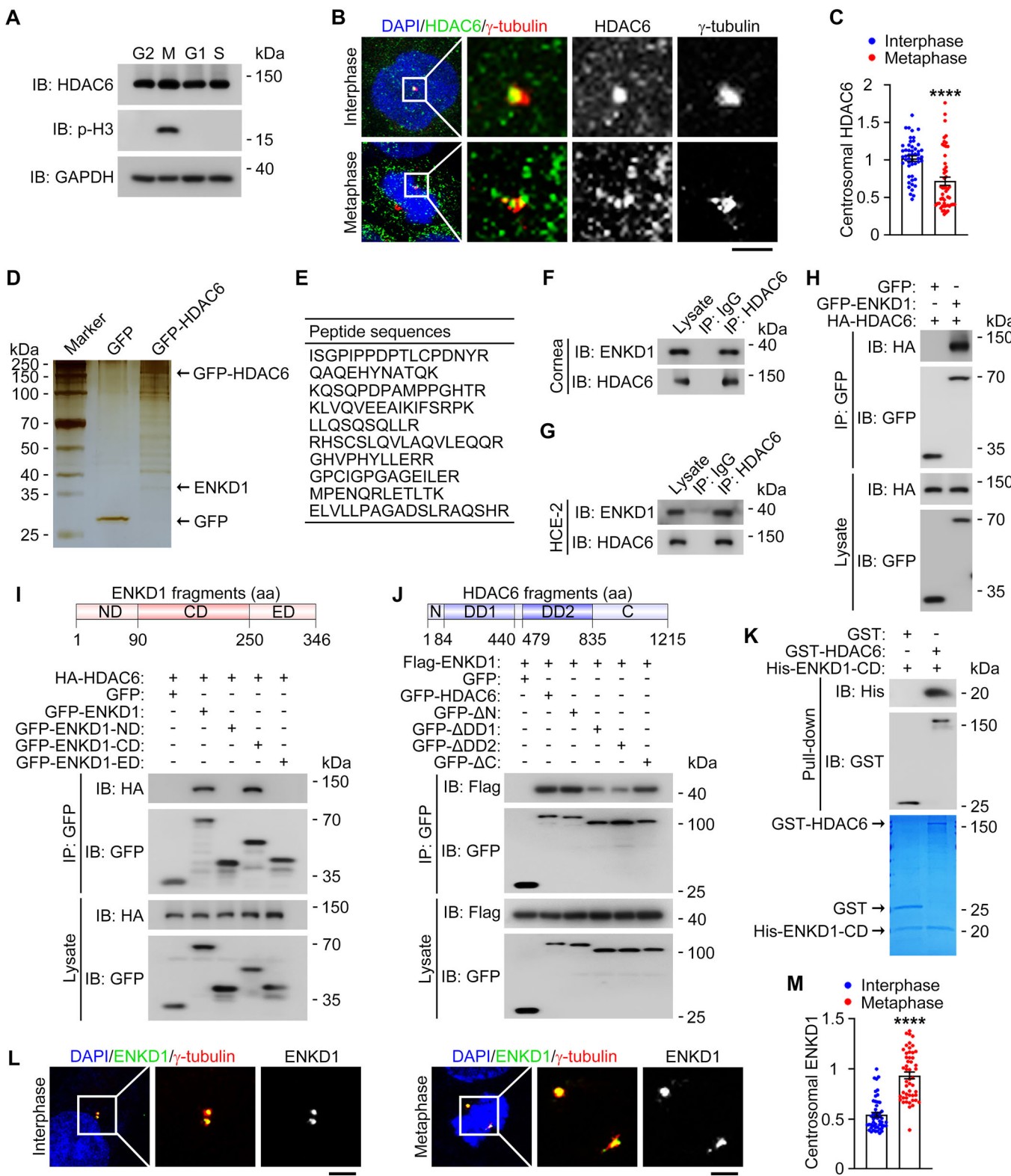

◀ **Figure 3. HDAC6 interacts with ENKD1 both in vivo and in vitro.**

(A) Immunoblot analysis of HDAC6, phosphorylated histone H3 (p-H3), and GAPDH in G1, S, G2, and M phases of HCE-2 cells. (B, C) Immunofluorescence images (B) and quantification of HDAC6 intensity (C, $n = 50$ cells from three independent experiments) in interphase and metaphase HCE-2 cells, and stained with antibodies against HDAC6 and γ-tubulin and DAPI. Scale bar, 1 μm. $P < 0.0001$. (D) Silver staining of proteins immunoprecipitated with GFP antibody from HCE-2 cells transfected with GFP-HDAC6 or GFP. (E) ENKD1 peptide sequences identified by mass spectrometry in the proteins immunoprecipitated from HCE-2 cells with GFP antibody. (F, G) Immunoprecipitation and immunoblotting showing the interaction between endogenous HDAC6 and ENKD1 in corneas (F) and HCE-2 cells (G). (H) Immunoprecipitation and immunoblotting showing the interaction of HA-HDAC6 with GFP-ENKD1 in HEK293T cells. (I) Schematic diagram of ENKD1 and identification of the domains of ENKD1 mediating its interaction with HDAC6. ND N-terminal domain. CD centrosomal domain, ED enkurin domain. (J) Schematic diagram of HDAC6 and identification of the domains of HDAC6 mediating its interaction with ENKD1. DD deacetylase domain. (K) GST pull-down showing the interaction of purified GST-HDAC6 with purified His-ENKD1-CD. (L, M) Immunofluorescence images (L) and quantification of ENKD1 intensity (M, $n = 50$ cells from three independent experiments) in interphase and metaphase HCE-2 cells, and stained with antibodies against ENKD1 and γ-tubulin and DAPI. Scale bar, 1 μm. $P < 0.0001$. Data are presented as mean ± SEM. Unpaired two-tailed Student's $t$ test was performed. ****$P < 0.0001$. Source data are available online for this figure.

then used the I-TASSER software to predict the three-dimensional structure of ENKD1; K98 was found to locate on the surface of ENKD1, and the K98R mutation appeared to alter the side chain at this specific site (Fig. EV4E).

We subsequently investigated whether HDAC6-mediated deacetylation of ENKD1 impacts its protein level or centrosomal localization. We found that HDAC6 did not affect the protein level of ENKD1 (Fig. EV4F). The centrosomal localization of ENKD1 diminished significantly upon overexpression of wild-type HDAC6 or its H215A mutant in HDAC6-depleted HCE-2 cells. In contrast, in HDAC6-depleted cells expressing the H610A mutant or the H215/H610A mutant, we observed little or no obvious changes in the centrosomal localization of ENKD1 (Fig. 4I,J). These findings suggest that the centrosomal localization ENKD1 is remarkably affected by its deacetylation by HDAC6. To verify this finding, we compared the centrosomal localization of three different forms of ENKD1, including wild-type ENKD1, the K98R mutant, and the K98Q mutant (substitution of the lysine 98 residue with glutamine) in corneal epithelial cells. The K98Q mutant exhibited a significant increase in centrosomal localization, while the K98R mutant showed a weakened centrosomal localization, in comparison to wild-type ENKD1 (Fig. 4K,L). Collectively, these results suggest that HDAC6-mediated deacetylation of ENKD1 modulates its centrosomal localization.

## ENKD1 acetylation at K98 is required for proper spindle orientation and positioning in corneal epithelial cells

The dramatic effect of HDAC6-mediated deacetylation on the centrosomal localization of ENKD1 suggests that this might be attributed to the activity of HDAC6 to regulate spindle orientation and positioning. To test this possibility, HCE-2 cells were transfected with two different ENKD1 siRNAs. Consistent with the previous findings (Zhong et al, 2022), knockdown of ENKD1 expression resulted in spindle misorientation without affecting spindle length (Fig. 5A,B; Appendix Fig. S4A–C). Furthermore, overexpression of wild-type ENKD1 or the K98Q mutant could rescue spindle orientation defects caused by ENKD1 siRNAs, whereas the K98R mutant failed to rescue (Fig. 5C–G; Appendix Fig. S4D–F). Additionally, we found that ENKD1 deficiency impaired spindle positioning without affecting cell diameter (Fig. 5H–J). Wild-type ENKD1 and the K98Q mutant, but not the K98R mutant, could rescue the improper spindle positioning induced by ENKD1 deficiency (Fig. 5K–M). Taken together, these data suggest that ENKD1 acetylation at K98 is required for proper spindle orientation and positioning in corneal epithelial cells.

## HDAC6-mediated deacetylation of ENKD1 reduces its interaction with γ-tubulin

We then explored how HDAC6-mediated deacetylation of ENKD1 impacts its centrosomal localization. We immunoprecipitated ENKD1 from HCE-2 cells and then performed mass spectrometry to identify proteins present in the ENKD1 immunoprecipitate. This analysis revealed the centrosomal protein γ-tubulin as an ENKD1-binding partner (Fig. 6A,B). Additional immunoprecipitation experiments demonstrated an interaction between endogenous ENKD1 and γ-tubulin both in the corneal tissue and in HCE-2 cells (Fig. 6C,D).

γ-Tubulin is known to associate with a number of centrosomal proteins to participate in diverse cellular processes (Ali et al, 2023; Vinopal et al, 2023). To better understand the relationship between γ-tubulin and ENKD1 at the centrosome/spindle pole, we transfected HCE-2 cells with two different γ-tubulin siRNAs. We found a significant attenuation in the localization of ENKD1 at the centrosome/spindle pole, following the knockdown of γ-tubulin expression (Fig. 6E–G). In parallel, we also evaluated the effect of ENKD1 depletion on the localization of γ-tubulin, and found that γ-tubulin localization at the centrosome/spindle pole was not affected after the knockdown of ENKD1 (Figs. 6H and EV5A). These findings indicate that γ-tubulin plays a crucial role in the centrosomal localization of ENKD1.

To investigate whether HDAC6-mediated deacetylation of ENKD1 affects its interaction with γ-tubulin, we transfected cells with wild-type or mutant ENKD1 and performed immunoprecipitation assays. We found that the K98Q mutation notably enhanced the interaction of ENKD1 with γ-tubulin; in contrast, the K98R mutation caused a weakened interaction of ENKD1 with γ-tubulin (Fig. 6I,J). To corroborate this finding, we examined the interaction between ENKD1 and γ-tubulin upon HDAC6 overexpression. We found that over-expression of HDAC6 wild-type or its H215A mutant, but not the H610A or H215/H610A mutant, largely impeded the interaction between ENKD1 and γ-tubulin (Fig. 6K,L). Additionally, we observed a weakened interaction between ENKD1 and γ-tubulin in corneas from mice injected with HDAC6 adenoviruses (Fig. EV5B,C). Collectively, these results suggest that ENKD1 acetylation promotes its interaction with γ-tubulin and thereby enhances its centrosomal localization.

## Discussion

The corneal epithelial layer acts as the principal barrier to various environmental infiltrates, including physical and chemical injuries

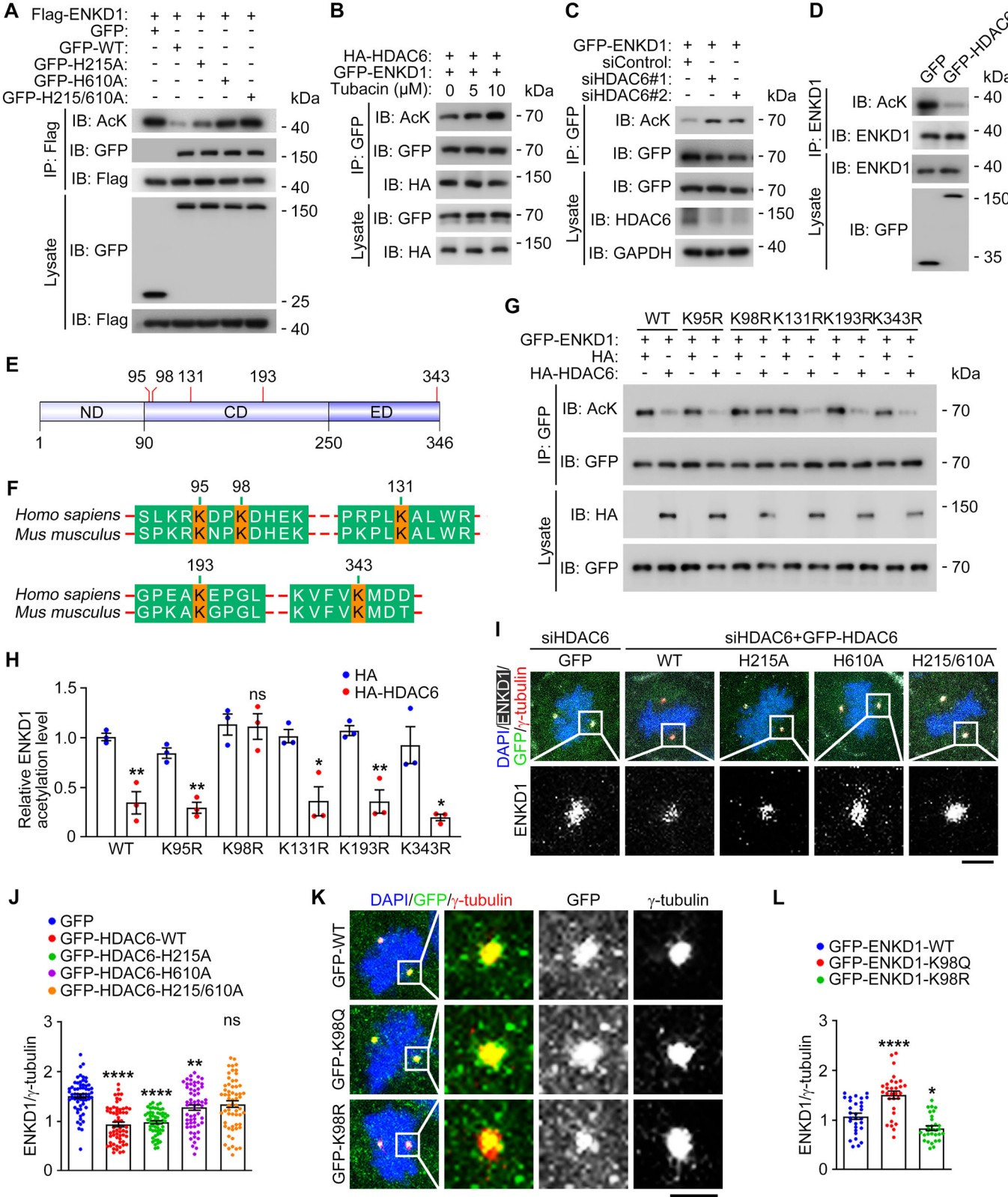

**Figure 4. HDAC6-mediated deacetylation of ENKD1 reduces its centrosomal localization.**

(A) Analysis of ENKD1 acetylation in HEK293T cells transfected with GFP, GFP-HDAC6 wild-type, GFP-HDAC6-H215A, GFP-HDAC6-H610A, or GFP-HDAC6-H215/610A, together with Flag-ENKD1. AcK, acetylated-lysine. (B) Analysis of ENKD1 acetylation in HEK293T cells transfected with GFP-ENKD1 and HA-HDAC6 and treated with different concentrations of tubacin. (C) Analysis of ENKD1 acetylation in HEK293T cells transfected with control or HDAC6 siRNAs, together with GFP-ENKD1. (D) Analysis of ENKD1 acetylation by immunoprecipitation with the ENKD1 antibody followed by immunoblotting with the acetylated-lysine antibody in corneal tissues, which were injected with control or HDAC6 adenoviruses. (E) ENKD1 acetylation sites predicted by the CSS-Palm software. (F) Alignment of the amino acid sequences of human and mouse ENKD1 proteins. The predicted acetylation sites are marked in orange. (G, H) HEK293T cells were transfected with HA or HA-HDAC6, together with GFP-ENKD1 wild-type or mutants. Immunoprecipitation and immunoblotting were then performed (G), and the relative ENKD1 acetylation level was quantified by densitometry (H, $n = 3$ independent experiments). The intensity of each ENKD1 acetylation band was normalized to that of the GFP band. WT, $P = 0.005$; K95R, $P = 0.002$; K98R, $P = 0.9113$; K131R, $P = 0.015$; K193R, $P = 0.0049$; K343R, $P = 0.0181$. (I, J) Immunofluorescence images (I) and quantification of ENKD1 intensity (J, $n = 60$ cells from three independent experiments) in HCE-2 cells transfected with HDAC6 siRNAs and GFP, GFP-HDAC6 wild-type, H215A, H610A, or H215/610A, and stained with antibodies against ENKD1 and γ-tubulin and DAPI. Scale bar, 1 μm. GFP-HDAC6-WT, $P < 0.0001$; GFP-HDAC6-H215A, $P < 0.0001$; GFP-HDAC6-H610A, $P = 0.0042$; GFP-HDAC6-H215/610A, $P = 0.0714$. (K, L) Immunofluorescence images (K) and quantification of ENKD1 intensity (L, $n = 30$ cells from three independent experiments) in HCE-2 cells transfected with GFP-ENKD1 wild-type, K98Q, or K98R, and stained with antibody against γ-tubulin and DAPI. Scale bar, 1 μm. GFP-ENKD1-K98Q, $P < 0.0001$; GFP-ENKD1-K98R, $P = 0.0172$. Data are presented as mean ± SEM. Non-parametric one-way ANOVA with post hoc analysis was performed for (J, L). Unpaired two-tailed Student's $t$ test was performed for (H). $*P < 0.05$; $**P < 0.01$; $****P < 0.0001$; ns not significant. Source data are available online for this figure.

and microbial infection (Di Girolamo and Park, 2023). In response to the environmental challenge, the corneal epithelium possesses a remarkable capacity for perpetual self-renewal and expedited wound healing for the maintenance of the corneal integrity (Altshuler et al, 2021; Lin et al, 2023). This capacity relies on the well-coordinated proliferation, migration, and differentiation of epithelial cells (Li et al, 2021). However, the molecular mechanisms that govern corneal epithelial homeostasis remain largely elusive. The present work identifies HDAC6 as a critical protein modulating spindle orientation and positioning in corneal epithelial cells. In addition, our study underscores a crucial role of HDAC6 in the deacetylation of ENKD1 to regulate spindle behavior in the corneal epithelium. These findings shed light on the molecular details of how epithelial homeostasis is maintained in the cornea.

Over the past decades, HDAC6 has been implicated in diverse biological processes (Magupalli et al, 2020; Osseni et al, 2022). Notably, HDAC6 reduces HIV-induced microtubule acetylation for effective viral invasion and replication (Elliott and O'Hare, 1998; Naranatt et al, 2005; Valenzuela-Fernández et al, 2005). Moreover, the deacetylation of microtubules by HDAC6 plays a vital role in cell movement (Li et al, 2011), immune synapse assembly (Serrador et al, 2004), intracellular transport (Naren et al, 2023), and ciliary homeostasis (Ran et al, 2020). Our present study uncovers a significant role for HDAC6 in the regulation of spindle behavior, furthering the importance of HDAC6 in microtubule-associated processes. Complementing these insights, previous investigations, including our own, have shown the localization of HDAC6 at the centrosome/basal body during the initial phase of cilium formation, with its subsequent relocation to the ciliary axoneme upon cilium disassembly (Prodromou et al, 2012; Ran et al, 2015). Building upon these findings, our current work reveals that as the spindle forms, the centrosomal localization of HDAC6 gradually diminishes. These observations suggest that the localization of HDAC6 undergoes dynamic changes to orchestrate the spatial and functional aspects of cellular infrastructure. Furthermore, the precise centrosomal localization of HDAC6, as well as its spatiotemporal dynamics during cell cycle progression, remain to be fully elucidated. Unraveling these details will provide critical insights into the molecular network at the centrosome and its regulatory mechanisms underlying the spindle behavior.

ENKD1 has been demonstrated to promote spindle orientation in basal keratinocytes for proper epidermal stratification (Zhong et al, 2022). In addition, the localization of this protein at the basal body to

drive ciliogenesis through the removal of CP110 from the mother centriole (Song et al, 2022a). ENKD1 also plays a part in the regulation of tumor development, including in non-small cell lung cancer and diffuse large B-cell lymphoma (Song et al, 2023; Song et al, 2022b). However, it remains unknown whether ENKD1 undergoes post-translational modifications. It is also unclear how the localization of this protein is regulated in biological processes. Our current study demonstrates that the centrosomal localization of ENKD1 is tightly controlled by HDAC6-mediated deacetylation to control spindle behavior in corneal epithelial cells. Our data also reveal that dysregulation of ENKD1 acetylation and localization results in corneal epithelial thickening, implicating a potential involvement of the HDAC6-ENKD1 axis in corneal epithelial pathologies.

The proper orientation and positioning of the mitotic spindle are critical for the accurate segregation of the genetic material and the maintenance of tissue architecture (Lechler and Mapelli, 2021). The centrosome, as the major microtubule-organizing center, plays a pivotal role in spindle assembly and orientation (Hoffmann, 2021). Our results indicate that the acetylation of ENKD1, modulated by HDAC6, is essential for maintaining its interaction with γ-tubulin. The ENKD1/γ-tubulin interaction is crucial for the ability of ENKD1 to anchor at the centrosome and exert its function in spindle orientation and positioning, which in turn ensures proper cell division orientation and corneal homeostasis. However, the specific molecular mechanisms by which ENKD1 functions at the centrosome to regulate spindle behavior remain to be fully elucidated. ENKD1 may act as an important regulator to facilitate the nucleation and organization of microtubules, ensuring the correct alignment of the spindle. To gain a deeper understanding of these mechanisms, it will be necessary to analyze whether the microtubule-regulating capacity of ENKD1 is dependent on its centrosomal localization. Additionally, it will be important to dissect the precise molecular interactions involved in the above processes, including the binding affinity and structural changes of ENKD1 upon deacetylation.

Disruption of corneal epithelial homeostasis has been implicated in various corneal diseases, which are among the most prevalent causes of blindness worldwide (Griffith et al, 2016; Price et al, 2021). Several novel strategies for the management of corneal diseases have emerged over the past decades, including stem cell therapy (Armitage et al, 2019; Ljubimov and Saghizadeh, 2015). However, there are very few effective pharmacological interventions for corneal diseases. Recent

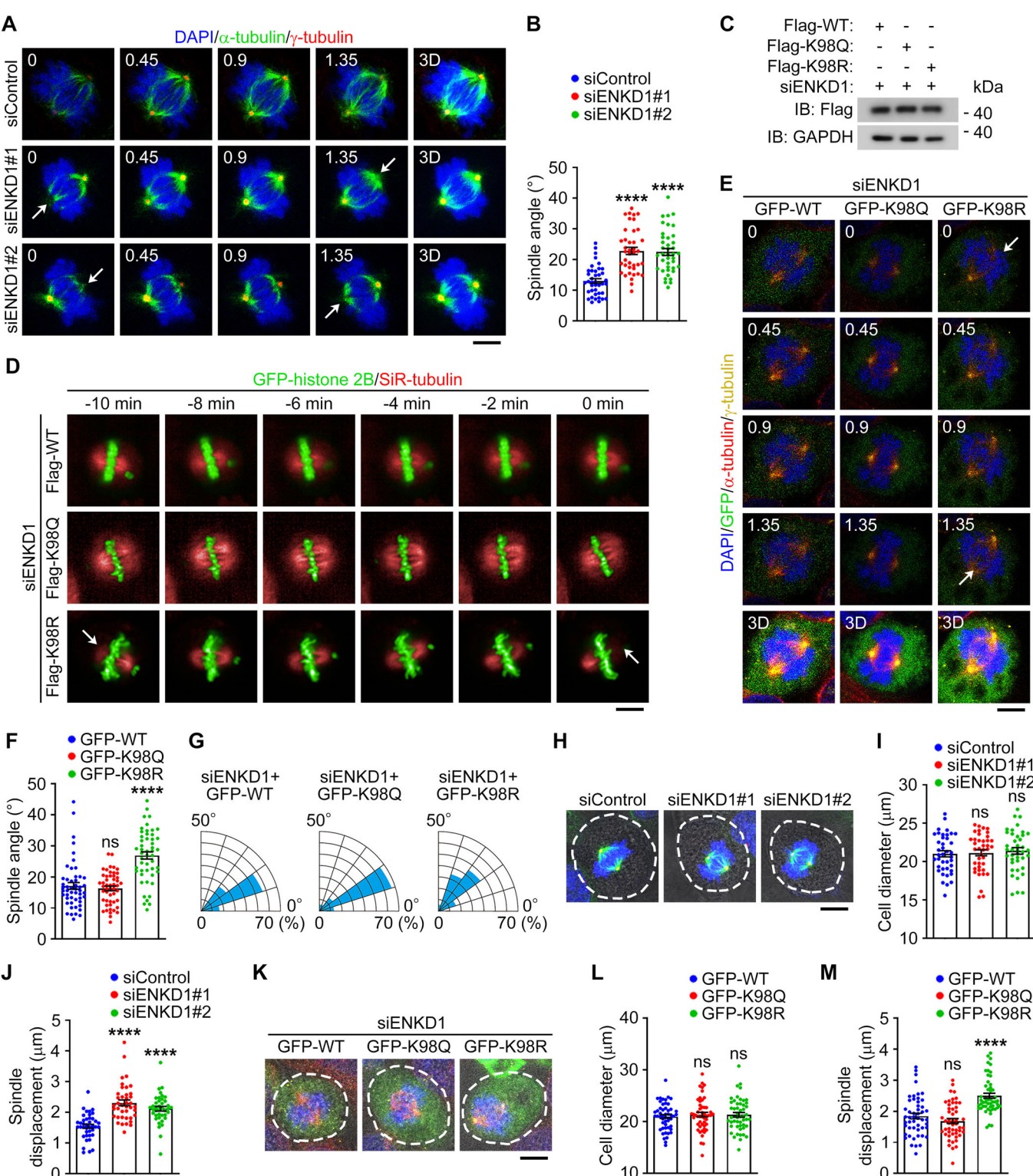

**Figure legend** (panels A–M as shown)

investigations, including our own studies, have demonstrated that HDAC6 is upregulated in various ocular diseases and that the use of HDAC6 inhibitors, as well as depletion of HDAC6, can effectively alleviate or prevent the associated pathological phenotypes of these diseases (Abouhish et al, 2020; Ran et al, 2020; Ran et al, 2022; Ran and

Zhou, 2020). These accumulating results, along with our finding that HDAC6 overexpression thickens the corneal epithelium and disturbs corneal homeostasis, suggest that targeting HDAC6 may hold promise for the management of ocular diseases, including keratoconus and keratitis.

◄ **Figure 5. ENKD1 acetylation at K98 is required for proper spindle orientation and positioning.**

(A, B) Immunofluorescence images (A) and quantification of spindle angle (B, $n = 40$ images from three independent experiments) in metaphase HCE-2 cells treated with control or ENKD1 siRNAs, and stained with antibodies against α-tubulin and γ-tubulin and DAPI. The white arrow refers to the spindle pole that does not appear on the corresponding focal plane (A). Scale bar, 4 μm. siENKD1#1, $P < 0.0001$; siENKD1#2, $P < 0.0001$. (C) Immunoblot analysis of Flag and GAPDH in HeLa cells transfected with ENKD1 siRNAs and Flag-ENKD1 wild-type, K98Q, or K98R. (D) Time-lapse images in mitotic HeLa cells transfected with ENKD1 siRNAs, GFP-histone 2B and Flag-ENKD1 wild-type, K98Q, or K98R. Microtubules were stained with SiR-tubulin (red). Anaphase onset was set at 0 min. The white arrow refers to the spindle pole that does not appear on the corresponding focal plane. Scale bar, 6 μm. (E–G) Immunofluorescence images (E), quantifications of spindle angle (F, $n = 50$ images from three independent experiments) and spindle angle distribution (G, $n = 50$ images from three independent experiments) in metaphase HCE-2 cells treated with ENKD1 siRNAs, together with GFP-ENKD1 wild-type, K98Q, or K98R, and stained with antibodies against α-tubulin and γ-tubulin and DAPI. The white arrow refers to the spindle pole that does not appear on the corresponding focal plane (E). Scale bar, 4 μm. GFP-K98Q, $P = 0.79$; GFP-K98R, $P < 0.0001$. (H–J) Immunofluorescence/bright-field images (H) and quantifications of cell diameter (I, $n = 40$ images from three independent experiments) and spindle displacement distance (J, $n = 40$ images from three independent experiments) in metaphase HCE-2 cells treated with control or ENKD1 siRNAs, and stained with antibodies against α-tubulin and γ-tubulin and DAPI. The white circle indicates the cell boundary (H). Scale bar, 5 μm. Cell diameter: siENKD1#1, $P = 0.9564$; siENKD1#2, $P = 0.7345$. Spindle displacement: siENKD1#1, $P < 0.0001$; siENKD1#2, $P < 0.0001$. (K–M) Immunofluorescence/bright-field images (K) and quantifications of cell diameter (L, $n = 50$ images from three independent experiments) and spindle displacement distance (M, $n = 50$ images from three independent experiments) in metaphase HCE-2 cells treated with ENKD1 siRNAs, together with GFP-ENKD1 wild-type, K98Q, or K98R, and stained with antibodies against α-tubulin and γ-tubulin and DAPI. The white circle indicates the cell boundary (K). Scale bar, 5 μm. Cell diameter: GFP-K98Q, $P = 0.7572$; GFP-K98R, $P = 0.7908$. Spindle displacement: GFP-K98Q, $P = 0.3101$; GFP-K98R, $P < 0.0001$. Data are presented as mean ± SEM. Non-parametric one-way ANOVA with post hoc analysis was performed. ****$P < 0.0001$; ns not significant. Source data are available online for this figure.

# Methods

### Reagents and tools table

| Reagent/resource | Reference or source | Identifier or catalog number |
| --- | --- | --- |
| **Experimental models** | | |
| HEK293T | ATCC | CRL-3216 |
| HeLa | ATCC | CRM-CCL-2 |
| HCE-2 | ATCC | CRL-3582 |
| **Recombinant DNA** | | |
| GFP-HDAC6 and its mutants | This paper | N/A |
| GFP-ENKD1 and its mutants | This paper | N/A |
| Flag-ENKD1 and its mutants | This paper | N/A |
| HA-HDAC6 | This paper | N/A |
| GST-HDAC6 | This paper | N/A |
| His-ENKD1-CD | This paper | N/A |
| Adenoviruses encoding GFP-HDAC6 or GFP vector | This paper | N/A |
| **Antibodies** | | |
| Rabbit anti-ENKD1 (IF) | Abcam | ab224560 |
| Rabbit anti-ENKD1 (WB) | Sigma-Aldrich | HPA041478 |
| Rabbit anti-HDAC6 (WB) | Millipore | 07-732 |
| Rabbit anti-HDAC6 (IF) | Abcam | ab239362 |
| Mouse anti-α-tubulin | Abcam | ab7291 |
| Mouse anti-centrin | Millipore | 04-1624 |

| Reagent/resource | Reference or source | Identifier or catalog number |
| --- | --- | --- |
| Mouse anti-acetylated α-tubulin | Sigma-Aldrich | T6793 |
| Rabbit anti-CDK10 | Proteintech | 30061-1-AP |
| Rabbit anti-β-catenin | Proteintech | 51067-2-AP |
| Rabbit anti-γ-tubulin | Sigma-Aldrich | T3320 |
| Mouse anti-GAPDH | Abways | AB0038 |
| Rabbit anti-GFP | Abways | AB0045 |
| Mouse anti-HA | Abways | AB0004 |
| Mouse anti-Flag | Abways | AB0008 |
| Rabbit anti-β-actin | Abways | AB0035 |
| Mouse anti-GST | Abways | AB0003 |
| Mouse anti-ZO-1 | Thermo Fisher | 33-9100 |
| Mouse anti-His tag | Abways | AB0002 |
| Mouse anti-acetylated-lysine | PTM BIO | PTM-105RM |
| Alexa Fluor 488-conjugated chicken anti-mouse secondary antibody | Thermo Fisher | A21200 |
| Alexa Fluor 488-conjugated donkey anti-rabbit secondary antibody | Thermo Fisher | A21206 |
| Alexa Fluor 568-conjugated donkey anti-mouse secondary antibody | Thermo Fisher | A10037 |

| Reagent/ resource | Reference or source | Identifier or catalog number |
|---|---|---|
| Alexa Fluor 568-conjugated donkey anti-rabbit secondary antibody | Thermo Fisher | A10042 |
| Alexa Fluor 647-conjugated donkey anti-mouse secondary antibody | Thermo Fisher | A31571 |
| Alexa Fluor 647-conjugated donkey anti-rabbit secondary antibody | Thermo Fisher | A31573 |
| **Oligonucleotides and other sequence-based reagents** | | |
| *Il-6* Forward | Beijing Genomics Institution (BGI) | TAGTCCTTCCTACCCCAATTTC |
| *Il-6* Reverse | BGI | TTGGTCCTTAGCCACTCCTTC |
| *Il-10* Forward | BGI | GCTCTTACTGACTGGCATGAG |
| *Il-10* Reverse | BGI | CGCAGCTCTAGGAGCATGTG |
| *Il-1β* Forward | BGI | GCAACTGTTCCTGAACTCAACT |
| *Il-1β* Reverse | BGI | ATCTTTTGGGGTCCGTCAACT |
| *Cdk10* Forward | BGI | TGGTCATGGGTTACTGCGAA |
| *Cdk10* Reverse | BGI | CCACGAAGCACCTGTAGCAT |
| *Gapdh* Forward | BGI | CTCAGGAGAGTGTTTCCTCGTC |
| *Gapdh* Reverse | BGI | ATGGGCTTCCCGTTGATGAC |
| siControl | GenePharma | CGUACGCGGAAUACUUCGA |
| siENKD1#1 | GenePharma | GUGGACUUCAUUCGUCACATT |
| siENKD1#2 | GenePharma | GGCCCAAAGUCUUCGUGAATT |
| siHDAC6#1 | GenePharma | GGAGUUAACUGGCAGGCAU |
| siHDAC6#2 | GenePharma | GCAGUUAAAUGAAUUCCAU |
| siy-tubulin#1 | GenePharma | CAGACGAUGAGCACUACAUTT |
| siy-tubulin#2 | GenePharma | CUGAAUGACAGGUAUCCUATT |
| **Chemicals, enzymes and other reagents** | | |
| Penicillin-Streptomycin | NCM Biotech | C100C5 |
| Tubacin | MedChemExpress | HY-13428 |
| DMEM medium | VivaCell | C3113-0500 |
| FBS | Gibco | A5256701 |
| DAPI | Sigma-Aldrich | D9542 |
| GFP nanoab agarose beads | NuoyiBio | GNA-25-500 |
| anti-HA beads | Sigma-Aldrich | A2095 |
| anti-Flag beads | Abmart | M20018 |
| protein A/G beads | Thermo Scientific | 20421 |
| glutathione resins | GenScript | L00206 |
| Ni-NTA resins | Thermo Scientific | 88221 |

| Reagent/ resource | Reference or source | Identifier or catalog number |
|---|---|---|
| **Software** | | |
| GraphPad Prism 8.0 | https://www.graphpad.com/ | |
| Image J | https://imagej.nih.gov/ij/index.html | |
| I-TASSER | https://zhanggroup.org/I-TASSER/ | |
| DESeq2 | https://bioconductor.org/packages/release/bioc/html/DESeq2.html | |
| R | https://www.r-project.org/ | |
| Ggpolt2 | https://cran.r-project.org/web/packages/ggplot2/index.html | |
| Photoshop | https://www.adobe.com/products/photoshop.html | |
| **Other** | | |
| Leica SP8 confocal microscope | Leica | N/A |
| HT-7800 transmission electron microscope | Hitachi | N/A |
| High Content Screening | PerkinElmer | N/A |

## Mice

All mouse experiments were conducted in accordance with the guidelines of the Animal Care and Use Committee of Shandong Normal University (AEECSDNU2023007). C57BL/6J mice were purchased from Charles River Laboratories (Beijing, China). Mice were provided with unrestricted access to food and water, and kept under standard lighting conditions, following a 12-h on/12-h off cycle.

## Cells

HCE-2, HEK293T, and HeLa cells were obtained from American Type Culture Collection. HeLa cells stably expressing GFP-histone 2B were obtained as described previously (Zhong et al, 2022). All cells were cultured in DMEM medium supplemented with 10% FBS, penicillin, and streptomycin at 37 °C with 5% $CO_2$.

## Plasmids and siRNAs

Mammalian expression plasmids for GFP-ENKD1, GFP-HDAC6, Flag-ENKD1, and HA-HDAC6 were constructed by using the

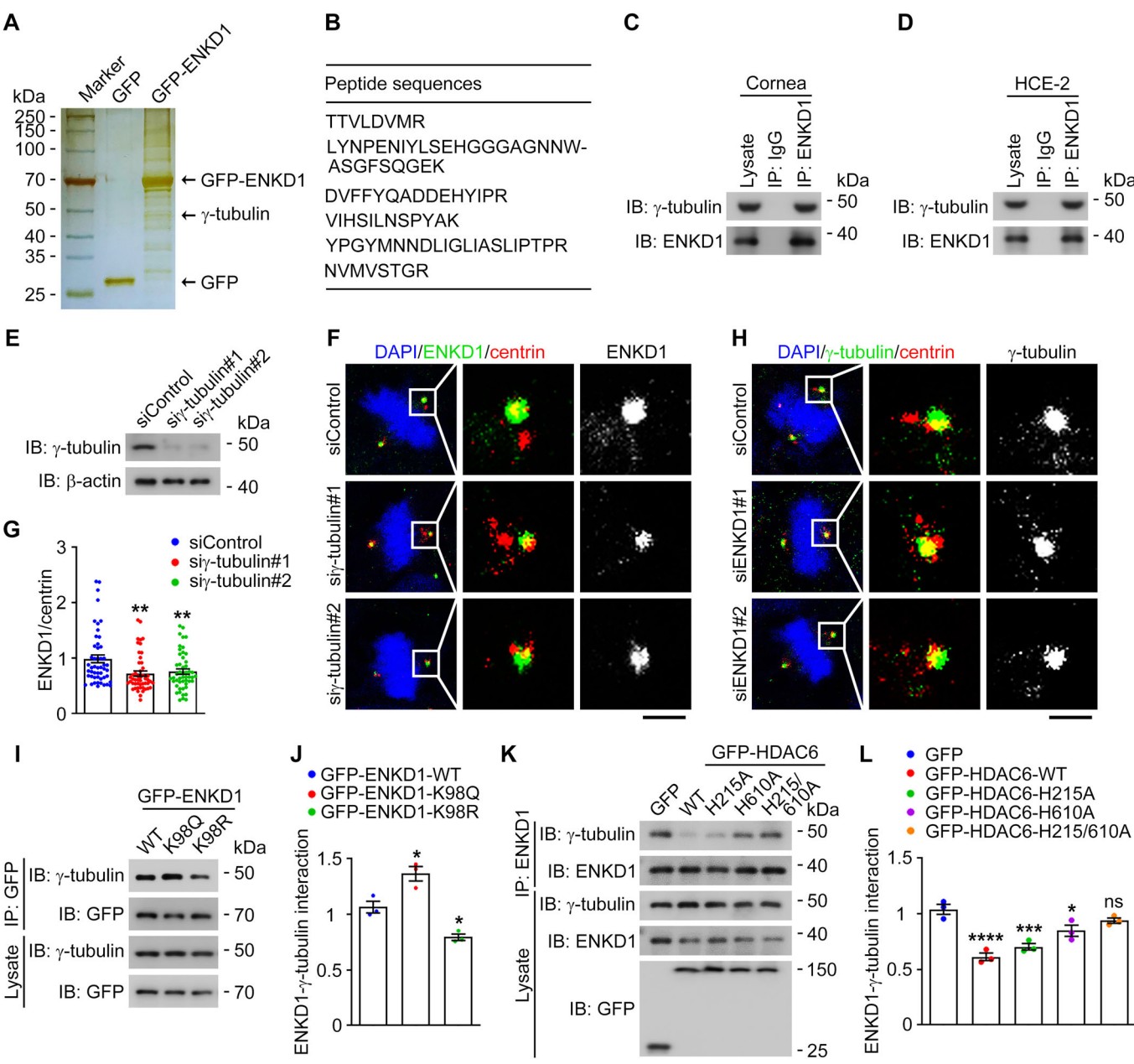

**Figure 6. HDAC6-mediated deacetylation of ENKD1 impedes its interaction with γ-tubulin.**

(A) Silver staining of proteins immunoprecipitated from HCE-2 cells transfected with GFP-ENKD1 or GFP. (B) γ-tubulin peptide sequences identified by mass spectrometry in the proteins immunoprecipitated from cells with the GFP antibody. (C, D) Immunoprecipitation and immunoblotting showing the interaction between endogenous ENKD1 and γ-tubulin. Immunoprecipitation was performed with the ENKD1 antibody or the corresponding control IgG in corneas (C) and HCE-2 cells (D). (E) Immunoblot analysis of γ-tubulin and β-actin in HCE-2 cells transfected with control or γ-tubulin siRNAs. (F, G) Immunofluorescence images (F) and quantification of ENKD1 intensity (G, $n = 50$ images from three independent experiments) in HCE-2 cells transfected with control or γ-tubulin siRNAs, and stained with antibodies against ENKD1 and centrin and DAPI. Scale bar, 1 μm. siγ-tubulin#1, $P = 0.0021$; siγ-tubulin#2, $P = 0.009$. (H) Immunofluorescence images in HCE-2 cells transfected with control or ENKD1 siRNAs, and stained with antibodies against γ-tubulin and centrin and DAPI. Scale bar, 1 μm. (I, J) Immunoprecipitation and immunoblotting (I) and quantification (J, $n = 3$ independent experiments) showing the ENKD1-γ-tubulin interaction in cells transfected with GFP-ENKD1 wild-type, K98Q, or K98R. The intensity of each γ-tubulin band was normalized to that of the GFP band. GFP-ENKD1-K98Q, $P = 0.0107$; GFP-ENKD1-K98R, $P = 0.0169$. (K, L) Immunoprecipitation and immunoblotting (K) and quantification (L, $n = 3$ independent experiments) showing the ENKD1-γ-tubulin interaction in cells transfected with GFP, GFP-HDAC6 wild-type, H215A, H610A, or H215/610A. The intensity of each γ-tubulin band was normalized to that of the ENKD1 band. GFP-HDAC6-WT, $P < 0.0001$; GFP-HDAC6-H215A, $P = 0.0003$; GFP-HDAC6-H610A, $P = 0.0158$; GFP-HDAC6-H215/610A, $P = 0.2393$. Data are presented as mean ± SEM. Non-parametric one-way ANOVA with post hoc analysis was performed. *$P < 0.05$; **$P < 0.01$; ***$P < 0.001$; ****$P < 0.0001$; ns not significant. Source data are available online for this figure.

pEGFP-C1, pcDNA3.1(+), and pCMV-C-HA vectors, respectively. The mammalian expression plasmids for GFP-α-tubulin and its mutants were described previously (Ran et al, 2015). Various truncated mutants were constructed by PCR. The siRNA-resistant forms of ENKD1 were constructed for rescue experiments.

## Cell transfection

Cells were transfected with plasmids by using Lipofectamine 3000 (Thermo Fisher Scientific), and transfected with siRNAs by using Lipofectamine RNAiMAX (Thermo Fisher Scientific). All transfections were performed in a serum-free culture condition following the manufacturer's instructions. Briefly, cells were seeded one day prior to transfection to achieve 30–50% confluency. These cells were cultured in DMEM supplemented with 10% FBS and without antibiotics. Following 16–24 h of cell seeding, the transfection mixture, which included plasmids/siRNAs and the transfection reagent, was added to the culture medium of the adherent cells. After 8 h post-transfection, the medium was replaced with fresh DMEM containing 10% FBS and cultured for additional 48 h.

## Intracameral injection of mice with adenoviruses

Adenoviruses encoding GFP-HDAC6 were generated using the pCMV-MCS-EGFP-SV40 vector (GeneChem, Shanghai, China). For adenovirus-mediated overexpression of HDAC6 in the cornea, male 8-week-old mice were used and anaesthetized with 2% isoflurane. One drop of 0.5% oxybuprocaine hydrochloride (Santen, Osaka, Japan) was applied to the surface of the cornea, and the iris was dilated using 0.5% tropicamide phenylephrine (Santen). Mice were then intracamerally injected with a solution containing $4 \times 10^{10}$ PFU mL$^{-1}$ adenoviruses, by using a 34-gauge needle/syringe (Hamilton, Reno, NV), and analyzed 10 days later.

## Mouse model of *P. aeruginosa* keratitis

*P. aeruginosa* (PAO1 strain) was obtained from Dr. Weihui Wu (Nankai University, Tianjin, China) and grown overnight in Luria-Bertani broth to the logarithmic phase (OD$_{600}$ of 1). The bacteria was then washed and resuspended in phosphate-buffered saline (PBS). C57BL/6 mice, aged 8–10 weeks, were anesthetized using a ketamine/xylazine solution, and the corneal epithelium was abraded with three parallel scratches using a sterile 26-gauge needle. Subsequently, 2 μL of the *P. aeruginosa* suspension (~$5 \times 10^4$ bacteria per eye) was applied topically. After 24 h, the mice were euthanized, and the corneas were imaged under the bright-field microscope to assess opacification.

## RNA preparation and RT-qPCR

Corneal samples were used for the isolation of total RNA following the manufacturer's protocol using the TRIzol reagent (15596018, Thermo Fisher Scientific). The isolated RNA was then subjected to cDNA synthesis using the M-MLV reverse transcriptase (M1701, Promega). RT-qPCR was performed using the LightCycler 480 II Real-time PCR system (Roche, Basel, Switzerland) with the FastStart Universal SYBR Green Master (04913914001, Roche), following the standard protocol. The expression levels of target genes were normalized to glyceraldehyde-3-phosphate dehydrogenase (GAPDH) to account for any sample-to-sample variation.

## Flow cytometry

For cell cycle analysis, HCE-2 cells were treated with adenoviruses encoding GFP or GFP-HDAC6 for 48 h. Subsequently, a cell cycle staining kit (CCS012, Multi Sciences, Hangzhou, China) was used for cell cycle analysis according to the manufacturer's instructions. Briefly, HCE-2 cells were centrifuged and washed twice with PBS. Cells were then resuspended in 1 mL of the DNA staining solution and 10 μL of the permeabilization solution, followed by incubation at room temperature for 30 min in the dark. Finally, the samples were analyzed using the Gallios flow cytometer (Beckman Coulter).

## EdU incorporation assay

For cell proliferation analysis, HCE-2 cells were treated with adenoviruses expressing GFP or GFP-HDAC6 adenoviruses for 48 h. The BeyoClick EdU cell proliferation kit with Alexa Fluor 555 (C0075S, Beyotime, Shanghai, China) was used to assess cell proliferation according to the manufacturer's instructions. Briefly, cells were incubated with the EdU buffer (10 μM) at room temperature for 2 h. The EdU buffer was then removed, and cells were fixed with 4% PFA for 15 min at room temperature. Cells were then washed with PBS and incubated with 0.3% Triton X-100 for 10 min to enhance permeabilization. Subsequently, cells were washed 3 times with PBS and incubated with click additive solution for 20 min. Finally, the reaction was terminated and incubated with DAPI.

## GST pull-down

GST-HDAC6 and His-ENKD1-CD (91-250 aa) proteins were expressed in *E. coli* and subsequently purified using either glutathione resins (GenScript, L00206) or Ni-NTA resins (Thermo Scientific, 88221). To perform GST pull-down, 100 μg of either GST or GST-HDAC6 fusion protein was immobilized onto 50 μL of glutathione resin in 1 mL of the lysis buffer and incubated with gentle rocking motion at 4 °C for 4–6 h. Beads were carefully washed three times using the lysis buffer, then incubated overnight at 4 °C with gentle rotation in the presence of 100 μg of His-ENKD1-CD protein. Beads were carefully washed six times using the in vitro binding buffer. The bound proteins were effectively eluted using the loading buffer and analyzed by immunoblotting.

## Histopathological analysis

Mouse eye tissues were immersed in 4% paraformaldehyde for 4 h, transferred to 70% ethanol, dehydrated through a serial alcohol gradient, and embedded in paraffin wax blocks. The samples were then sliced along the sagittal plane to prepare 6-μm sections for histological analysis. To measure corneal thickness, sections with the largest diameter were selected. The sections were then stained with the H&E solution (G1120, Solarbio). Images were captured and analyzed using a DM3000 microscope (Leica, Wetzlar).

## Immunoprecipitation and immunoblotting

Cells and tissues were lysed using a lysis buffer consisting of 50 mM Tris-HCl, 150 mM NaCl, 1 mM EDTA, 3% glycerinum, and 1% NP-40, supplemented with a protease inhibitor cocktail (Thermo Fisher Scientific). For immunoprecipitation, lysates were incubated with GFP nanoab agarose beads (NuoyiBio, GNA-25-500), anti-HA beads (Sigma-Aldrich, A2095), anti-Flag beads (Abmart, M20018), or protein A/G beads (Thermo Scientific, 20421) pre-incubated with primary antibodies overnight at 4 °C. Lysates were denatured, and the proteins were separated by SDS-PAGE. Immunoprecipitates were subjected to SDS-PAGE and then immunoblotted with antibodies. Protein bands were visualized by using the luminol reagent (Millipore).

## Fluorescence microscopy

Mouse eye tissues were cryo-embedded in Tissue-Tek OCT (Sakura) on dry-ice slabs. Frozen sections were cut into 10-μm slices and mounted on slides. Thin sections were fixed with 4% paraformaldehyde for 20 min, and permeabilized in 0.5% Triton X-100/PBS for 20 min. The tissues were then blocked in 4% bovine serum albumin (BSA) for 1 h and stained with primary antibodies at 4 °C overnight. They were then stained with secondary antibodies and DAPI (Sigma-Aldrich). Cells grown on coverslips were permeabilized with ice-cold methanol, and blocked with 4% BSA. Then, cells were sequentially probed with the primary antibodies, fluorophore-conjugated secondary antibodies, and DAPI. The fluorescence intensity of centrosomal proteins was measured with the Fiji software (National Institutes of Health). For live-cell imaging, cells transfected with GFP-histone 2B were seeded onto glass-bottom dishes and cultured to 30–40% confluency. Subsequently, cells were transfected with the plasmids and labeled with SiR-tubulin (Cytoskeleton, CY-SC002). Imaging was then performed using a laser confocal high-content imaging system (PerkinElmer) maintained at 37 °C in an environment containing 5% $CO_2$.

## Mass spectrometry

To identify HDAC6- or ENKD1-binding proteins, GFP-HDAC6 or GFP-ENKD1 was overexpressed in HEK293T cells and immuno-precipitated with anti-GFP beads, and the immunoprecipitated proteins were subjected to mass spectrometry by Applied Protein Technology. A Q-Exactive mass spectrometer (Thermo Fisher Scientific) and a gel-based LC-MS approach were used for the analysis.

## Transmission electron microscopy

Mouse eyes were dissected and fixed with 0.25% glutaraldehyde in 0.1 M sodium cacodylate for 24 h at 4 °C. After removal of the cornea, the tissues were fixed for additional 2 h and post-fixed in 1% osmium tetroxide for 1 h. The specimen was then dehydrated through a series of ethanol followed by resin infiltration. Corneas were then embedded in Spurr low viscosity resin and cured for 72 h at 65 °C. Ultrathin sections were sliced in 50 nm, stained with uranyl acetate and lead citrate, and examined with an HT-7800 transmission electron microscope (Hitachi) at 80 kV.

## Bioinformatics

Differential gene expression analyses between samples were performed using DESeq2 (Love et al, 2014). A heatmap was generated using the 'pheatmap' function from the R packages, and correlation coefficients were calculated using the 'cor' function in R (Version 4.1.1). A volcano plot was created using the 'ggplot2' package. For the identification of differentially expressed genes, the following thresholds were set: $P < 0.05$, log2FoldChange > 1, and log2FoldChange < −1.

## Statistical analysis

GraphPad Prism (GraphPad Software, La Jolla, CA) was used for statistical analysis. Results were shown as means ± SEM unless otherwise indicated. Significant differences between two groups were analyzed using the unpaired two-tailed Student's $t$ test. For multiple-condition comparisons, non-parametric one-way analysis of variance (ANOVA) test with Dunnett's multiple comparisons test was used.

## Ethical statement

All applicable institutional and/or national guidelines for the care and use of animals were followed. The use of mice was approved by the Animal Care and Use Committee of Shandong Normal University.

# Data availability

This study includes no data deposited in external repositories.

The source data of this paper are collected in the following database record: biostudies:S-SCDT-10_1038-S44319-025-00438-0.

# Peer review information

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

## Acknowledgements

We thank Ms. Heng Guo (Electron Microscopy Core, Shandong Normal University) for assistance in imaging and Dr. Weihui Wu (Nankai University) for providing *P. aeruginosa* PAO1. This work was supported by grants from the National Natural Science Foundation of China (32170829, 32300694, 32170687, 32241014, and 82470403), Henan Provincial Science and Technology Research Project (242102310113), and Henan Provincial Young and Middle-aged Health Science and Technology Innovation Talent Program (YQRC2024008).

## Author contributions

**Ting Song**: Data curation; Formal analysis; Investigation; Methodology; Project administration. **Xueqing Han**: Data curation; Formal analysis; Methodology. **Hanxiao Yin**: Formal analysis; Methodology. **Junkui Zhao**: Software; Methodology. **Mingming Ma**: Data curation; Software. **Xiaonuan Wen**: Data curation; Formal analysis; Methodology. **Chunli Liu**: Data curation; Formal analysis; Methodology. **Yiyang Yue**: Data curation; Formal analysis; Methodology. **Huijie Zhao**: Writing—review and editing. **Jun Zhou**: Supervision; Funding acquisition; Writing—review and editing. **Yang Yang**: Funding acquisition; Resources; Formal analysis; Supervision; Methodology. **Jie Ran**: Funding acquisition; Investigation; Project administration; Writing—review and editing. **Min Liu**: Supervision; Funding acquisition; Project administration; Writing—review and editing.

Source data underlying figure panels in this paper may have individual authorship assigned. Where available, figure panel/source data authorship is listed in the following database record: biostudies:S-SCDT-10_1038-S44319-025-00438-0.

## Disclosure and competing interests statement

The authors declare no competing interests.

# Expanded View Figures

**Figure EV1. Depletion of HDAC6 does not significantly affect the corneal epithelium.**

(A) Volcano plots showing the number of genes differentially regulated between healthy ($n = 3$ samples), KC ($n = 3$ samples), BK ($n = 3$ samples), and VK ($n = 3$ samples) corneas. Comparisons between the three different experimental groups are shown (healthy *vs.* KC, healthy *vs.* BK, and healthy *vs.* VK). The blue dots represent downregulated genes and the red dots represent upregulated genes. (B) Top 9 enriched terms in the GO analysis of differentially expressed genes listed in the three different experimental groups (healthy vs KC, healthy vs BK, healthy vs VK). (C–E) Quantification of *Il-6*, *Il-10*, and *Il-1β* mRNA levels by RT-qPCR in corneas from healthy or BK mice ($n = 3$ independent experiments). All data are normalized to *Gapdh* mRNA levels. *Il-6*, $P = 0.0004$; *Il-10*, $P = 0.006$; *Il-1β*, $P = 0.0006$. (F) Immunofluorescence images of the cornea from control or HDAC6 adenovirus-injected mice. Scale bar, 30 μm. (G, H) Immunoblot analysis of HDAC6, GFP, and GAPDH in corneas from control or HDAC6 adenovirus-injected mice. The red arrow indicates GFP-HDAC6, and the green arrow indicates endogenous HDAC6 (G). The relative levels of GFP-HDAC6 and endogenous HDAC6 were determined by densitometry (H, $n = 3$ independent experiments) (I, J) Quantification of the thickness of the stroma (I, $n = 30$ images from three independent experiments) and endothelium (J, $n = 30$ images from three independent experiments) from control or HDAC6 adenovirus-injected mice. Scale bar, 15 μm. (K) Schematic illustration of the strategy used for generating *Hdac6* knockout mice. Exons 10–13 of the mouse *Hdac6* gene were replaced by a vector containing neomycin (Neo) and zeocin (Zeo) cassettes. (L) Immunoblotting of HDAC6 and GAPDH in wild-type and *Hdac6* knockout mice. (M–P) H&E staining images (M) and quantification of the thickness of the epithelium (N, $n = 15$ images from three independent experiments), stroma (O, $n = 15$ images from three independent experiments) and endothelium (P, $n = 15$ images from three independent experiments) in wild-type and *Hdac6* knockout mice. Scale bar, 30 μm. Data are presented as mean ± SEM. Unpaired two-tailed Student's *t* test was performed for (C, D, E, I, J, N, O, P). **$P < 0.01$; ***$P < 0.001$; ns, not significant. Differential gene expression analysis was performed using the DESeq2 package in R for (A). Enrichment analysis was performed using Metascape (http://www.metascape.org/) for (B). Source data are available online for this figure.

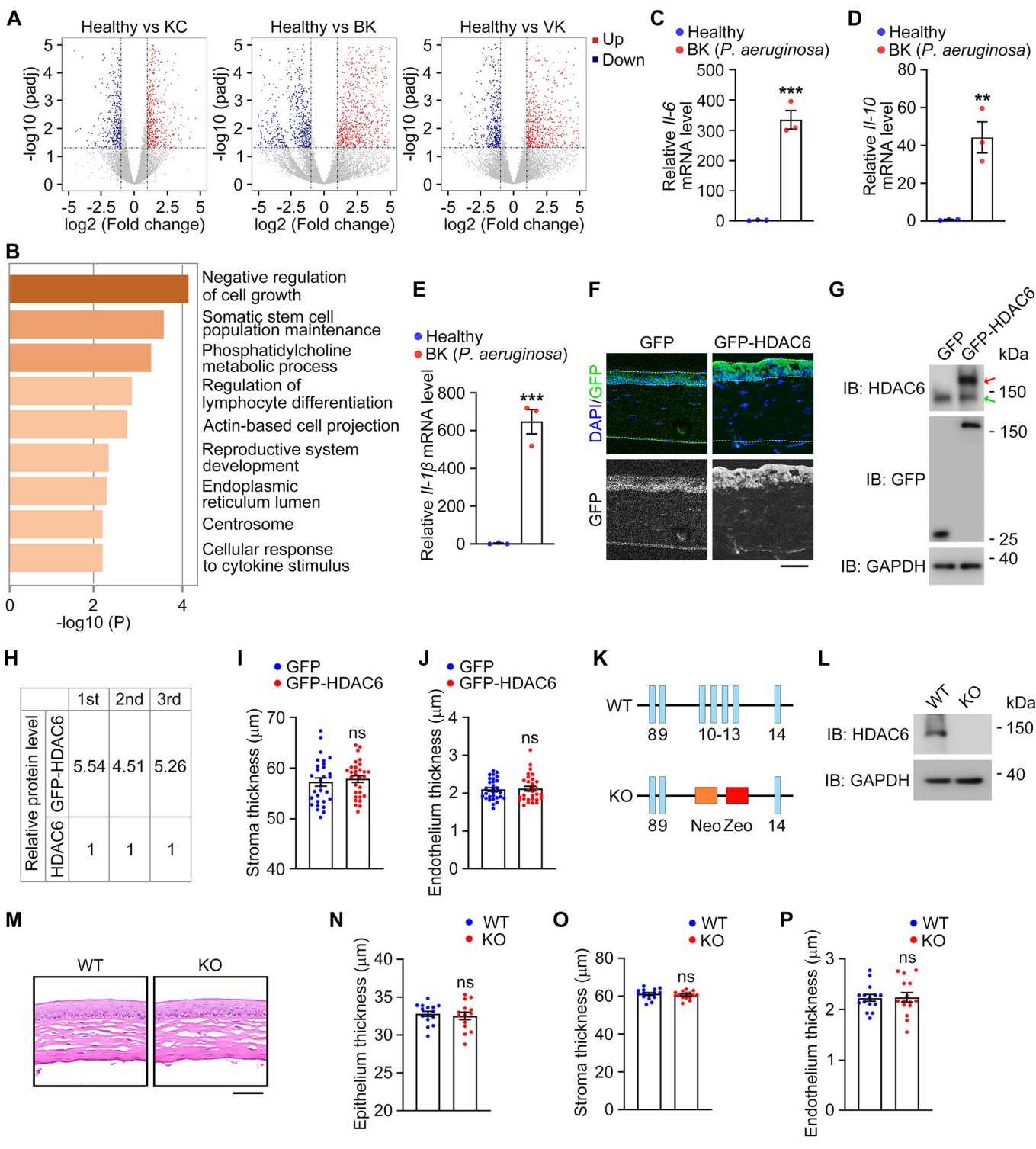

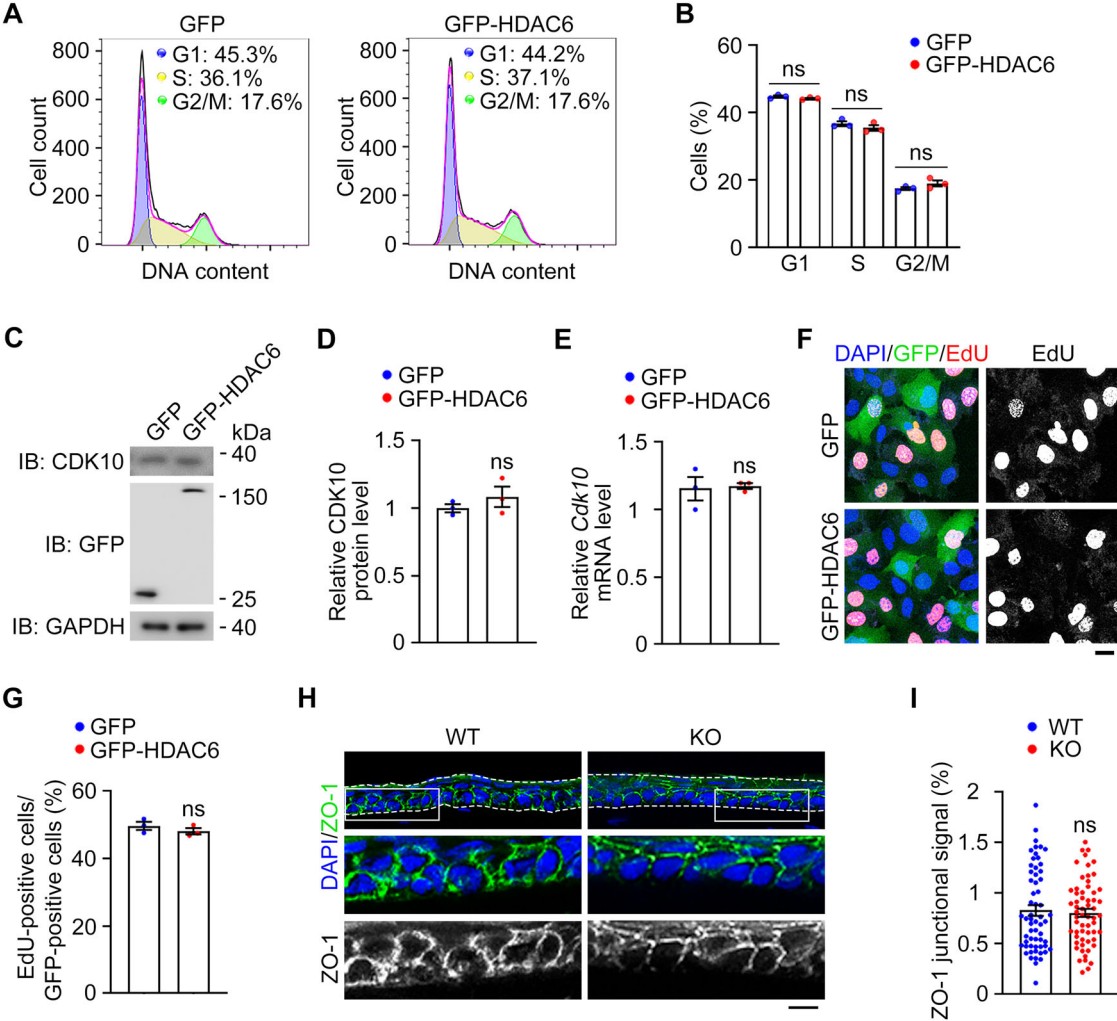

**Figure EV2. HDAC6 overexpression does not affect cell cycle progression or cell proliferation.**

(A, B) Flow cytometric analysis of the cell cycle (A) and quantification of the percentage of cells at G1, S, or G2/M phase (B, $n = 3$ independent experiments) for HCE-2 cells treated with GFP or GFP-HDAC6 adenoviruses. (C, D) Immunoblotting (C) and quantification (D, $n = 3$ independent experiments) showing the level of CDK10 in corneas from control or HDAC6 adenovirus-injected mice. The intensity of the CDK10 band was normalized to that of the GAPDH band. (E) Quantification of the *Cdk10* mRNA level in corneas from control or HDAC6 adenovirus-injected mice ($n = 3$ independent experiments). All data are normalized to the *Gapdh* mRNA level. (F, G) Immunofluorescence images (F) and quantification (G, $n = 3$ independent experiments) of the EdU-positive HCE-2 cells treated with GFP or GFP-HDAC6 adenoviruses. (H, I) Immunofluorescence images (H) and quantification of the ZO-1 junctional signals (I, $n = 60$ cells from three independent experiments) in wild-type and *Hdac6* knockout mice, stained with the ZO-1 antibody and DAPI. Scale bar, 30 μm. Data are presented as mean ± SEM. Unpaired two-tailed Student's *t* test was performed. ns not significant. Source data are available online for this figure.

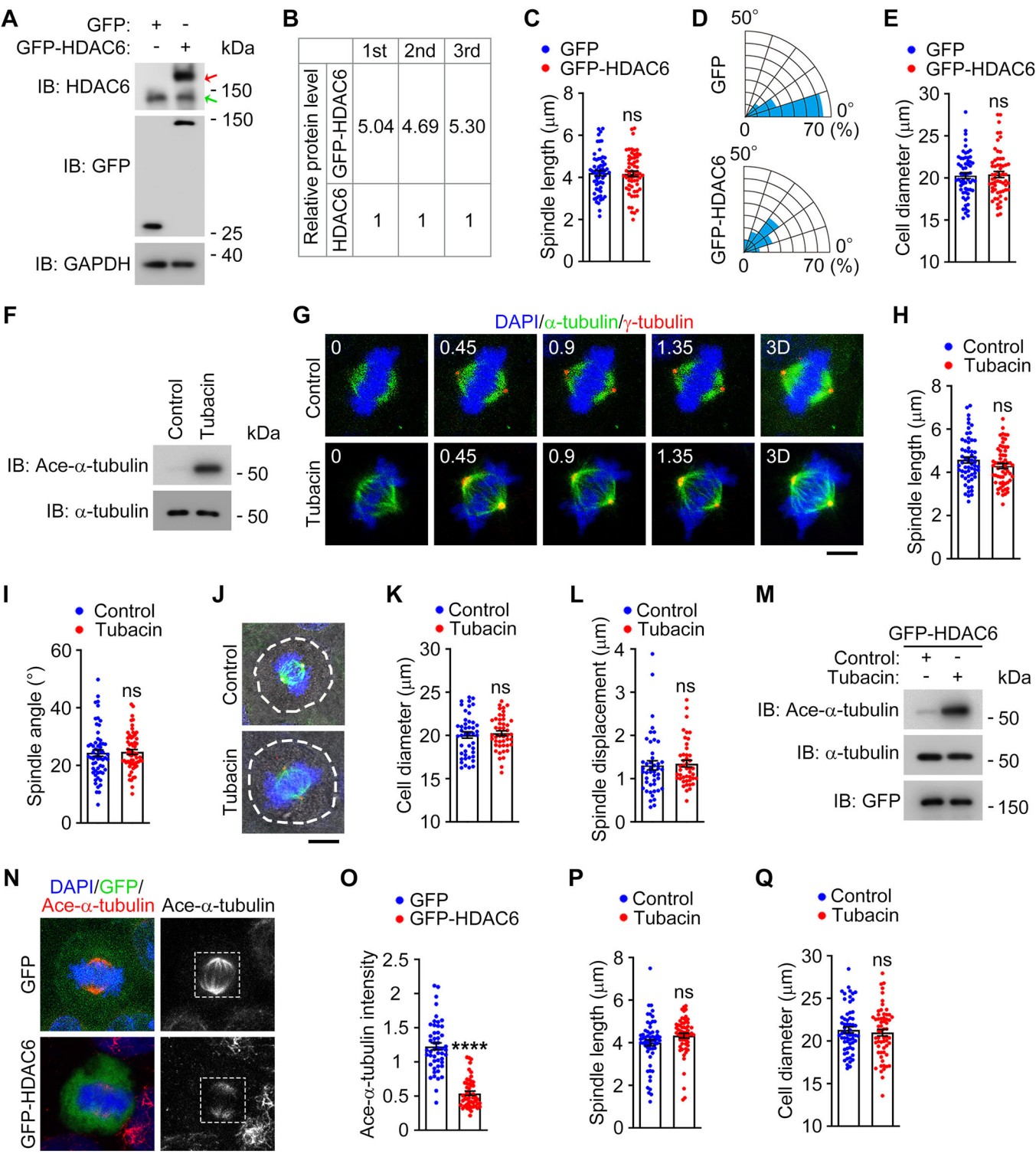

◀ **Figure EV3. Inhibition of HDAC6 does not affect spindle orientation or positioning.**

(A, B) Immunoblot analysis of HDAC6, GFP, and GAPDH in HCE-2 cells transfected with GFP-HDAC6 or GFP. The red arrow indicates GFP-HDAC6, and the green arrow indicates endogenous HDAC6 (A). The relative levels of GFP-HDAC6 and endogenous HDAC6 were determined by densitometry (B, $n = 3$ independent experiments). (C–E) Quantification of spindle length (C, $n = 60$ images from three independent experiments), spindle angle distribution (D, $n = 60$ images from three independent experiments), and cell diameter (E, $n = 60$ images from three independent experiments) of metaphase HCE-2 cells transfected with GFP-HDAC6 or GFP. (F) Immunoblot analysis of acetylated α-tubulin and α-tubulin in HCE-2 cells treated with DMSO or tubacin. (G–I) Immunofluorescence images (G) and quantifications of spindle length (H, n = 60 images from three independent experiments) and spindle angle (I, $n = 60$ images from three independent experiments) in metaphase HCE-2 cells treated with DMSO or tubacin, and stained with antibodies against α-tubulin and γ-tubulin and DAPI. Scale bar, 4 μm. (J–L) Immunofluorescence/bright-field images (J) and quantification of cell diameter (K, $n = 45$ images from three independent experiments) and spindle displacement distance (L, $n = 45$ images from three independent experiments) in metaphase HCE-2 cells treated with DMSO or tubacin, and stained with antibodies against α-tubulin and γ-tubulin and DAPI. The white circle indicates the cell boundary (J). Scale bar, 6 μm. (M) Immunoblot analysis of acetylated α-tubulin, α-tubulin and GFP in HCE-2 cells transfected with GFP-HDAC6 and treated with DMSO or tubacin. (N, O) Immunofluorescence images (N) and quantification of acetylated α-tubulin intensity (O, $n = 50$ images from three independent experiments) of metaphase HCE-2 cells transfected with GFP-HDAC6 or GFP, and stained with antibodies against acetylated α-tubulin and DAPI. The area inside the white box was used for quantitative analysis (N). Scale bar, 4 μm. $P < 0.0001$. (P, Q) Quantification of spindle length (P, $n = 60$ images from three independent experiments) and cell diameter (Q, $n = 60$ images from three independent experiments) from metaphase HCE-2 cells transfected with GFP-HDAC6 and treated with tubacin (10 μM). Data are presented as mean ± SEM. Unpaired two-tailed Student's $t$ test was performed. ****$P < 0.0001$; ns not significant. Source data are available online for this figure.

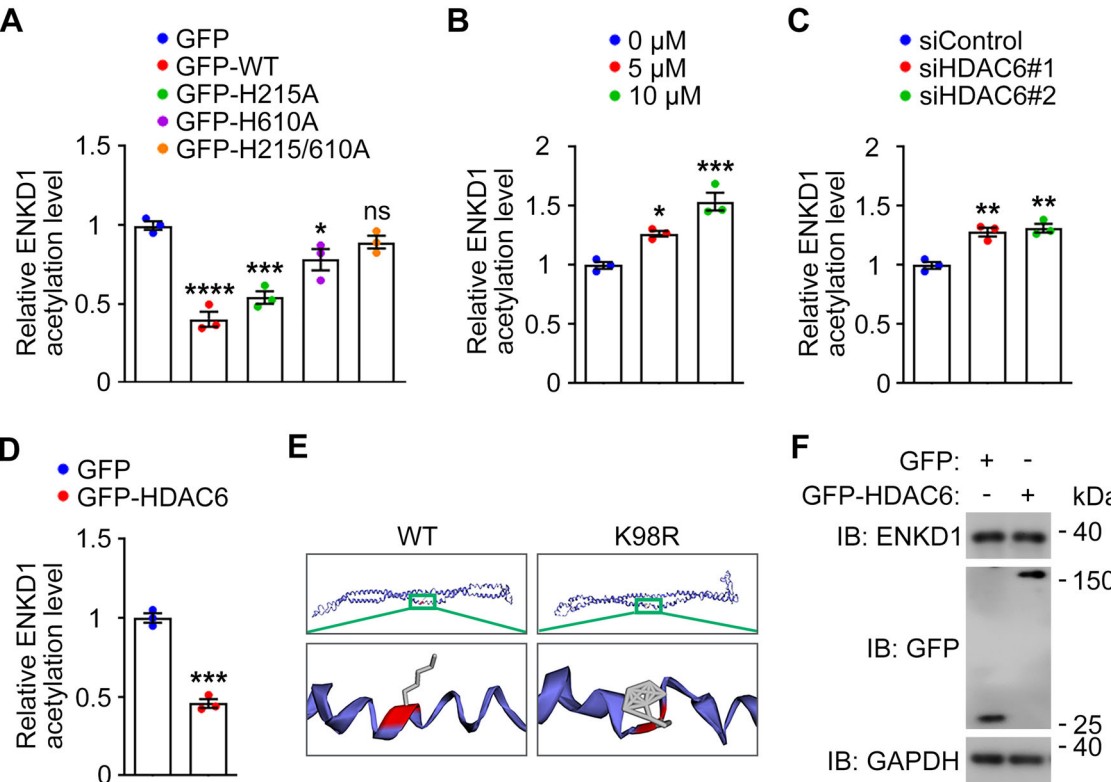

**Figure EV4.  HDAC6 mediates the deacetylation of ENKD1.**

(**A**) Quantification of the ENKD1 acetylation level ($n = 3$ independent experiments) in HEK293T cells transfected with GFP, GFP-HDAC6 wild-type, H215A, H610A, or H215/610A, together with Flag-ENKD1. GFP-WT, $P < 0.0001$; GFP-H215A, $P = 0.0002$; GFP-H610A, $P = 0.025$; GFP-H215/610A, $P = 0.3596$. (**B**) Quantification of the ENKD1 acetylation level ($n = 3$ independent experiments) in HEK293T cells transfected with HA-HDAC6 and GFP-ENKD1, and treated with different concentrations of tubacin. 5 μM, $P = 0.0149$; 10 μM, $P = 0.0004$. (**C**) Quantification of the ENKD1 acetylation level ($n = 3$ independent experiments) in HEK293T cells transfected with control or HDAC6 siRNAs, together with GFP-ENKD1. siHDAC6#1, $P = 0.0021$; siHDAC6#2, $P = 0.0011$. (**D**) Quantification of the ENKD1 acetylation level ($n = 3$ independent experiments) in corneal tissues injected with control or HDAC6 adenoviruses. $P = 0.0002$. (**E**) Three-dimensional structure of ENKD1 wild-type and K98R mutant predicted with the I-TASSER software. The red color represents the K98 residue, while the gray color represents the side chain. (**F**) Immunoblot analysis of ENKD1, GFP, and GAPDH in HCE-2 cells transfected with GFP or GFP-HDAC6. Data are presented as mean ± SEM. Non-parametric one-way ANOVA with post hoc analysis was performed for (**A–C**). Unpaired two-tailed Student's $t$ test was performed for (**D**). *$P < 0.05$; **$P < 0.01$; ***$P < 0.001$; ****$P < 0.0001$; ns not significant. Source data are available online for this figure.

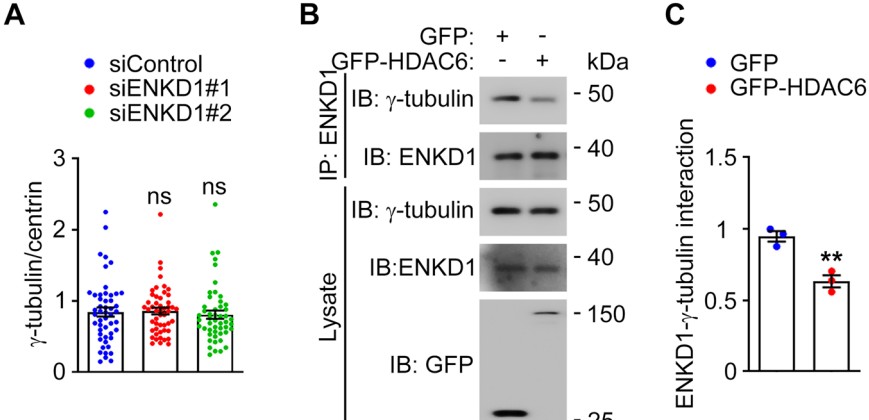

**Figure EV5.  HDAC6 inhibits the interaction of ENKD1 with γ-tubulin.**

(A) Quantification of γ-tubulin intensity (*n* = 50 images from three independent experiments) in HCE-2 cells transfected with control or ENKD1 siRNAs. (B, C) Immunoprecipitation and immunoblotting (B) and quantification (C, *n* = 3 independent experiments) showing the ENKD1-γ-tubulin interaction in the corneal tissues injected with control or HDAC6 adenoviruses. The intensity of each γ-tubulin band was normalized to that of the ENKD1 band. *P* = 0.0048. Data are presented as mean ± SEM. Non-parametric one-way ANOVA with post hoc analysis was performed for (A). Unpaired two-tailed Student's *t* test was performed for (C). **$P < 0.01$; ns not significant. Source data are available online for this figure.

