## [Peer Review File · EMBO Reports]

HDAC6 deacetylates ENKD1 to regulate mitotic spindle behavior and corneal epithelial homeostasis

Ting Song, Xueqing Han, Hanxiao Yin, Junkui Zhao, Mingming Ma, Xiaonuan Wen, Chunli Liu, Yiyang Yue, Huijie Zhao, Jun Zhou, Yang Yang, Jie Ran, and Min Liu

Corresponding authors: Min Liu (minliu@nankai.edu.cn), Jie Ran (jran@sdu.edu.cn), Yang Yang (fccyangyang430@zzu.edu.cn)

Review Timeline:

Submission Date:	24th Jul 24
Editorial Decision:	4th Oct 24
Revision Received:	2nd Jan 25
Editorial Decision:	5th Feb 25
Revision Received:	11th Feb 25
Accepted:	6th Mar 25

Editor: Deniz Senyilmaz Tiebe

Transaction Report:

Dear Dr. Liu,

Thank you for submitting your research manuscript to our journal, which was now seen by three referees, whose reports are copied below.

I apologize for the delay in getting back to you, it took longer than anticipated to receive the full set of referee reports.

Referees express interest in the proposed role of HDAC6 in regulation of mitotic spindle orientation and corneal epithelial homeostasis. However, they also raise significant concerns that need to be addressed to consider publication here. In particular,

- The proposed effect of HDAC6 overexpression on tight junctions (referee #1, major comment 2) and the spindle orientation (referee #1, major comment 3) requires additional support.
- More evidence is required to support the proposed regulation of ENKD1 centrosomal regulation by HDAC6 mediated deacetylation (referee #1, major comment 7).
- Whether HDAC6 mediated microtubule deacetylation plays a role in the observed spindle behavior needs to be tested (referee #2, point 1).
- Other comments regarding technical concerns and missing controls.

Should you be able to address these concerns satisfactorily, we would like to invite you to submit a revised manuscript. Please contact me if you have questions or comments regarding the revision and discuss the revision further (also by video chat).

We realize that it is difficult to revise to a specific deadline. In the interest of protecting the conceptual advance provided by the work, we recommend a revision within 3 months. Please discuss the revision progress ahead of this time with me if you require more time to complete the revisions, or if you have questions or comments regarding the revision (also by video chat).

1. A data availability section providing access to data deposited in public databases is missing (where applicable).
2. Your manuscript contains statistics and error bars based on $n=2$. Please use scatter plots in these cases.

You can submit the revision either as a Scientific Report or as a Research Article. For Scientific Reports, the revised manuscript can contain up to 5 main figures and 5 Expanded View figures, and it should not exceed 27000 characters. If the revision leads to a manuscript with more than 5 main figures it will be published as a Research Article. In this case the Results and Discussion section should be separate. If a Scientific Report is submitted, these sections have to be combined. This will help to shorten the manuscript text by eliminating some redundancy that is inevitable when discussing the same experiments twice. In either case, all materials and methods should be included in the main manuscript file.

4) a .docx formatted letter INCLUDING the reviewers' reports and your detailed point-by-point responses to their comments. As

part of the EMBO publication's Transparent Editorial Process, EMBO reports publishes online a Review Process File (RPF) to accompany accepted manuscripts. This File will be published in conjunction with your paper and will include the referee reports, your point-by-point response and all pertinent correspondence relating to the manuscript.

<https://www.embopress.org/page/journal/14693178/authorguide#transparentprocess>

5) a complete author checklist, which you can download from our author guidelines

<https://www.embopress.org/page/journal/14693178/authorguide>. Please insert information in the checklist that is also reflected in the manuscript. The completed author checklist will also be part of the RPF.

6) Please note that all corresponding authors are required to supply an ORCID ID for their name upon submission of a revised manuscript (<<https://orcid.org/>>). Please find instructions on how to link your ORCID ID to your account in our manuscript tracking system in our Author guidelines

<<https://www.embopress.org/page/journal/14693178/authorguide#authorshipguidelines>>

7) Before submitting your revision, primary datasets produced in this study need to be deposited in an appropriate public database (see <https://www.embopress.org/page/journal/14693178/authorguide#datadeposition>). Please remember to provide a reviewer password if the datasets are not yet public. The accession numbers and database should be listed in a formal "Data Availability" section placed after Materials & Method (see also

<https://www.embopress.org/page/journal/14693178/authorguide#datadeposition>). Please note that the Data Availability Section is restricted to new primary data that are part of this study. * Note - All links should resolve to a page where the data can be accessed. *

Additional information on source data and instruction on how to label the files are available:

<https://www.embopress.org/page/journal/14693178/authorguide#sourcedata>

9) Our journal encourages inclusion of *data citations in the reference list* to directly cite datasets that were re-used and obtained from public databases. Data citations in the article text are distinct from normal bibliographical citations and should directly link to the database records from which the data can be accessed. In the main text, data citations are formatted as follows: "Data ref: Smith et al, 2001" or "Data ref: NCBI Sequence Read Archive PRJNA342805, 2017". In the Reference list, data citations must be labeled with "[DATASET]". A data reference must provide the database name, accession number/identifiers and a resolvable link to the landing page from which the data can be accessed at the end of the reference. Further instructions are available at <http://www.embopress.org/page/journal/14693178/authorguide#referencesformat>

10) Regarding data quantification (see Figure Legends:

<https://www.embopress.org/page/journal/14693178/authorguide#figureformat>)

11) The journal requires a statement specifying whether or not authors have competing interests (defined as all potential or

actual interests that could be perceived to influence the presentation or interpretation of an article). In case of competing interests, this must be specified in your disclosure statement. Further information: <https://www.embopress.org/competing-interests>

12) Please also note our reference format:

13) All Materials and Methods need to be described in the main text using our 'Structured Methods' format, which is required for all research articles. According to this format, the Methods section includes a Reagents and Tools Table (listing key reagents, experimental models, software and relevant equipment and including their sources and relevant identifiers) followed by a Methods and Protocols section describing the methods using a step-by-step protocol format. The aim is to facilitate adoption of the methodologies across labs. More information on how to adhere to this format as well as a downloadable template (.docx) for the Reagents and Tools Table can be found in our author guidelines:

I look forward to seeing a revised version of your manuscript when it is ready. Please let me know if you have questions or comments regarding the revision.

Kind regards,

Deniz Senyilmaz Tiebe

Deniz Senyilmaz Tiebe, PhD
Senior Scientific Editor
EMBO Reports

Referee #1:

The proper positioning of the mitotic spindle is crucial for morphogenesis and development. In this manuscript, the authors propose that overexpression of HDAC6 affects spindle orientation in corneal epithelial tissue, as well as in the HCE-2 human corneal epithelial cell line. While the authors suggest a connection between HDAC6 function and disease biology, the experimental data provided do not consistently support the manuscript's claims. Several critical issues, such as a lack of appropriate controls, unclear interpretations of microscopy images, and a reliance on overexpression (OE) in the presence of endogenous protein, undermine the conclusions. Therefore, I cannot recommend the publication of this paper without the inclusion of additional controls to substantiate their findings. Below are specific comments for the authors to consider.

Major comments

-In Fig. 1, The authors express GFP-HDAC6 using adenovirus; however, the level of GFP-HDAC6 overexpression compared to the endogenous protein remains unclear. The authors should quantify the amount of GFP-HDAC6 relative to the endogenous protein, as this is key to correlating HDAC6 upregulation with disease conditions. Similarly, in Fig. 2A, it would be useful to assess HDAC6 overexpression in HCE-2 cells in comparison to endogenous levels.

-In Fig. 1G, the authors conclude that the HDAC6 overexpression severely disrupts the tight junction. However, it appears that the overall morphology of the cells is affected. HDAC6 OE cells look much more elongated, and the impact on the barrier might be secondary. The total number of cells seems to increase upon HDAC6 OE condition. One of the cell cycle genes that were upregulated in the microarray dataset was CDK10; it could well be that HDAC6 OE leads to a change in the gene expression of many cell cycle/proliferation-related genes, leading to the observed phenotype. The authors should test this possibility in their setup.

-Given the apparent increase in cell number upon HDAC6 OE, the authors should verify whether the spindle orientation observed in fixed tissue is due to cell misorientation rather than spindle misorientation. In Fig. 1L, for example, the cell appears misaligned, which could naturally lead to a misaligned spindle. Therefore, I am not convinced that HDAC6 OE directly causes spindle misorientation, and further validation is required to confirm this finding in tissue, before analyzing this phenotype in cell culture model.

-In Fig. S3, The authors use Tubacin to inhibit HDAC6 activity, reporting no significant effect on spindle orientation. This finding should be reinforced by knocking down HDAC6, specifically in HCE-2 cells. Additionally, while the authors observed increased acetylated tubulin levels via Western blot after Tubacin treatment, they should perform immunofluorescence analysis of acetylated tubulin in mitotic cells, comparing the amount in cells overexpressing HDAC6 with those where HDAC6 is knocked down.

-All spindle orientation assays performed in Fig. 2Q and 2S are done in a setting where the endogenous HDAC6 is present. Therefore, it is unclear what the implication of endogenous protein is on such a mutant version of HDAC6. The authors should check the relevance of this HDAC6 mutant on spindle orientation in the absence of endogenous protein.

-The authors claim that HDAC6 OE affects spindle orientation and therefore analyzed endogenous HDAC6 localization in Fig. 3, and they uncovered that spindle pole localization of HDAC6 is decreases during mitosis. In lines 196-199, they suggest that these observations relate to downstream centrosomal proteins, but this rationale is unclear, especially since the loss of HDAC6 activity does not result in a spindle orientation phenotype. The explanation should be based on HDAC6 localization following overexpression.

-In Fig. 4, the authors claim that HDAC6 OE deacetylates ENKD1 at the centrosome, affecting its localization. However, ENKD1 remains centrosome-enriched in control cells despite significant levels of HDAC6 during mitosis (Fig. 3). It is unclear why endogenous HDAC6 would not affect ENKD1 localization but does so upon overexpression conditions, especially when the amount of HDAC6 overexpression is unclear. Also, most in vivo experiments were conducted where HDAC6 wild-type or mutant versions were expressed in the presence of endogenous protein, complicating the interpretation. Therefore, the authors should perform such experiments with proper controls to bolster their conclusions.

Minor comments

-Why does the nuclear size appear larger in cells overexpressing GFP-HDAC6 (Fig. 1E)?

-To analyze the impact of HDAC6 OE on spindle orientation, authors should also perform live-imaging analysis to follow the behavior of the mitotic spindle. Similarly, it would be informative to know the spindle behaviors where ENKD1 wild-type and mutant version affect spindle angle (Fig. 5).

Referee #2:

Manuscript by Song et al. compared the transcriptomic profiles of healthy and pathological corneas, and discovered that HDAC6 was dramatically upregulated in corneal diseases. They next showed that overexpression of HDAC6 in mice led to thickening of the corneal epithelium by impairing the spindle orientation and positioning in corneal epithelial cells. Further study validated that HDAC6 deacetylated ENKD1 at K98 to compromise its interaction with γ -tubulin, thus impacting the centrosomal localization of ENKD1 and its regulation in spindle dynamics. Overall, the study is well conducted with several interesting findings. Before considering this manuscript for publication, there are some points that the authors should address.

1. One major concern is that previous studies have reported HDAC6 could directly regulate microtubule stability via deacetylation of alpha-tubulin. Thus, authors are suggested to clarify whether the affected spindle behavior in HDAC6-overexpressed corneal epithelial cells is involved in the deacetylation of microtubules by HDAC6.
2. Fluorescence images of HDAC6-GFP are encouraged to provide for verification of HDAC6 localization at spindle poles in corneal epithelial cells.
3. Authors are recommended to further clarify in the Discussion why weakened interaction of ENKD1 with γ -tubulin would result in the defective spindle orientation and positioning.
4. Some English grammar and wording errors need to be revised.

Referee #3:

The manuscript by Song et al examines the role of the deacetylase HDAC6 in regulating mitotic spindle behavior and corneal epithelial homeostasis. In a nutshell, the authors show that HDAC6 deacetylates the protein ENKD1, which impairs its interaction with γ -tubulin and centrosomal localisation. This thus reveals a critical role for HDAC6 in the control of corneal epithelial homeostasis.

In general the experiments are well executed and convincing. However, a number of points should be addressed before I can recommend the paper for publication. Below are specific points to address.

Figure 1: Based on publicly available transcriptomics data the authors identify overexpression of HDAC6 in corneal disease samples. This observation is the basis for their subsequent experiments. Could the authors obtain some samples from corneal diseases and show by immunoblot that HDAC6 is overexpressed in these samples? This would strengthen the paper.

Figure 3D: This shows a silver staining of proteins immunoprecipitated by a GFP antibody. Why is the antibody not visible? Were the proteins eluted from beads with stably bound antibody. This is not explained, also not in methods.

Fig. 3F and G: The immunoprecipitation does not appear to work, as HDAC6 is not visibly enriched in the IP vs the lysate. Can the authors comment and explain in the text?

Fig. 4B: This panel is a bit perplexing. It is shown that GFP-ENKD1 co-IP very well with HDAC6; this is unusual, as typically it is not possible to immunoprecipitate the substrates of an enzyme (or only very weakly if at all). Furthermore, when tubacin is added -which blocks the catalytic pocket of HDAC6- it does not decrease the co-IP; also that is strange, as one would expect that then there is less co-IP. Can the authors comment on this?

General comments: 1. quantification of the immunoblots (including mentioning how many times the experiments have been done) would be useful to appreciate the magnitude of the effects observed.

2. the methods section is not detailed enough; as it is now it is not possible to reproduce the experiments.

Point-to-Point Responses to the Reviewers' Critiques

EMBOR-2024-60066V1

"HDAC6 deacetylates ENKD1 to regulate mitotic spindle behavior and corneal epithelial homeostasis"

We sincerely appreciate the thorough analyses and constructive suggestions provided by the reviewers, which have been very helpful in guiding us to improve our manuscript. After reading the enclosed point-to-point responses, we hope that the editor and the reviewers will concur with us that we have addressed all the raised concerns in a satisfactory manner.

Reviewer #1 (Remarks to the Author):

..... While the authors suggest a connection between HDAC6 function and disease biology, the experimental data provided do not consistently support the manuscript's claims. Several critical issues, such as a lack of appropriate controls, unclear interpretations of microscopy images, and a reliance on overexpression (OE) in the presence of endogenous protein, undermine the conclusions. Therefore, I cannot recommend the publication of this paper without the inclusion of additional controls to substantiate their findings. Below are specific comments for the authors to consider.

Response: Thank you very much for your thorough and insightful review of our manuscript. We greatly appreciate the time and effort you have invested in providing us with detailed comments and suggestions. We are grateful for the opportunity to revise our manuscript and are committed to ensuring that the experimental data and interpretations are robust and well-supported. The following are our point-to-point responses.

Major comments

1. In Fig. 1, The authors express GFP-HDAC6 using adenovirus; however, the level of GFP-HDAC6 overexpression compared to the endogenous protein remains unclear. The authors should quantify the amount of GFP-HDAC6 relative to the endogenous protein, as this is key to correlating HDAC6 upregulation with disease conditions. Similarly, in Fig. 2A, it would be useful to assess HDAC6 overexpression in HCE-2 cells in comparison to endogenous levels.

Response: We appreciate the insightful comment from the reviewer. To address the concern regarding the quantification of GFP-HDAC6 overexpression relative to endogenous HDAC6, we have performed immunoblotting using mouse corneas injected with GFP-HDAC6 adenoviruses and HCE-2 cells transfected with GFP-HDAC6. Our results showed that GFP-HDAC6 levels were approximately 5 times higher than those of the endogenous protein in both the cornea (Fig. EV1G,H)

and HCE-2 cells (Fig. EV3A,B). Additionally, we established a bacterial keratitis (BK) mouse model. Immunoblotting revealed that HDAC6 protein levels in BK mice were also about 5 times higher than in the control group (Fig. 1F,G).

2. In Fig. 1G, the authors conclude that the HDAC6 overexpression severely disrupts the tight junction. However, it appears that the overall morphology of the cells is affected. HDAC6 OE cells look much more elongated, and the impact on the barrier might be secondary. The total number of cells seems to increase upon HDAC6 OE condition. One of the cell cycle genes that were upregulated in the microarray dataset was CDK10; it could well be that HDAC6 OE leads to a change in the gene expression of many cell cycle/proliferation-related genes, leading to the observed phenotype. The authors should test this possibility in their setup.

Response: Thank you for this great question. Indeed, HDAC6 plays a crucial role in maintaining tissue homeostasis through the regulation of various substrate proteins. It is plausible that HDAC6 overexpression might affect tight junctions indirectly by influencing cell cycle or proliferation-related genes. To address this possibility, we examined the effect of HDAC6 on the cell cycle progression. We found that HDAC6 overexpression did not alter the distribution of cells in G1, S, and G2/M phases (Fig. EV2A,B). We also investigated the expression of CDK10 in the corneas of mice injected with HDAC6 adenoviruses and found no significant changes in either protein or mRNA levels compared to control adenovirus-injected mice (Fig. EV2C-E). Additionally, we analyzed the EdU-positive cells in HDAC6-overexpressing HCE-2 cells, and our results demonstrated that HDAC6 overexpression had no obvious effect on the proliferation of corneal epithelial cells (Fig. EV2F,G).

3. Given the apparent increase in cell number upon HDAC6 OE, the authors should verify whether the spindle orientation observed in fixed tissue is due to cell misorientation rather than spindle misorientation. In Fig. 1L, for example, the cell appears misaligned, which could naturally lead to a misaligned spindle. Therefore, I am not convinced that HDAC6 OE directly causes spindle misorientation, and further validation is required to confirm this finding in tissue, before analyzing this phenotype in cell culture model.

Response: We sincerely apologize for the quality of images and lack of detailed explanations. In the revised manuscript, we have included more representative images to demonstrate the orientation of corneal epithelial cells in the corneal tissue. H&E-staining of corneal epithelial sections revealed that control corneal epithelial cells exhibited symmetric division with their orientation parallel to the basement membrane. In contrast, corneal epithelial cells treated with HDAC6 adenovirus showed a division direction deviating from the basement membrane (Fig. 1P). We have described this in detail in the manuscript and figure legends.

4. In Fig. S3, the authors use Tubacin to inhibit HDAC6 activity, reporting no significant effect on spindle orientation. This finding should be reinforced by knocking down HDAC6, specifically in HCE-2 cells. Additionally, while the authors observed

increased acetylated tubulin levels via Western blot after Tubacin treatment, they should perform immunofluorescence analysis of acetylated tubulin in mitotic cells, comparing the amount in cells overexpressing HDAC6 with those where HDAC6 is knocked down.

Response: We greatly appreciate your detailed suggestions. Following your advice, we used siRNAs targeting HDAC6 to treat HCE-2 cells and found that HDAC6 knockdown did not affect the spindle behavior (Appendix Fig. S1D-I). Additionally, we performed immunofluorescence analysis of acetylated tubulin in mitotic cells. Our results showed that the intensity of acetylated tubulin was significantly decreased in cells overexpressing HDAC6 (Fig. EV3N,O) and significantly increased in cells where HDAC6 was knocked down (Appendix Fig. S1A-C), which are consistent with our immunoblotting results.

5. All spindle orientation assays performed in Fig. 2Q and 2S are done in a setting where the endogenous HDAC6 is present. Therefore, it is unclear what the implication of endogenous protein is on such a mutant version of HDAC6. The authors should check the relevance of this HDAC6 mutant on spindle orientation in the absence of endogenous protein.

Response: Thanks for pointing this out. We apologize for not having set up the appropriate controls. Following your suggestion, we overexpressed HDAC6 wild-type and the relevant mutants in cells where endogenous HDAC6 was knocked down. Our results showed that overexpression of HDAC6 wild-type or the H215A mutant impaired spindle orientation and positioning, while overexpression of the H610A or H215/H610A mutants did not (Fig. 2N-Q and Appendix Fig. S2A-C).

6. The authors claim that HDAC6 OE affects spindle orientation and therefore analyzed endogenous HDAC6 localization in Fig. 3, and they uncovered that spindle pole localization of HDAC6 is decreases during mitosis. In lines 196-199, they suggest that these observations relate to downstream centrosomal proteins, but this rationale is unclear, especially since the loss of HDAC6 activity does not result in a spindle orientation phenotype. The explanation should be based on HDAC6 localization following overexpression.

Response: Thank you very much for your valuable suggestions. We also apologize for the premature conclusions we drew and for the lack of detailed and clear explanations in our original manuscript. In response to your feedback, we have revised our manuscript and examined the localization of GFP-HDAC6 during mitosis. Our data showed that the localization of GFP-HDAC6 at the spindle pole was also reduced during mitosis, but this change was relatively modest compared to the alteration in endogenous HDAC6 localization (Appendix Fig. S3A,B). This difference is likely due to the much higher expression of exogenous HDAC6 compared to endogenous HDAC6 (Fig. EV3A,B). These results suggest that the reduced localization of HDAC6 at the centrosome is crucial for maintaining proper spindle behavior, and that overexpression of HDAC6 may interfere with this dynamic localization and lead to abnormal spindle behavior.

7. In Fig. 4, the authors claim that HDAC6 OE deacetylates ENKD1 at the centrosome, affecting its localization. However, ENKD1 remains centrosome-enriched in control cells despite significant levels of HDAC6 during mitosis (Fig. 3). It is unclear why endogenous HDAC6 would not affect ENKD1 localization but does so upon overexpression conditions, especially when the amount of HDAC6 overexpression is unclear. Also, most in vivo experiments were conducted where HDAC6 wild-type or mutant versions were expressed in the presence of endogenous protein, complicating the interpretation. Therefore, the authors should perform such experiments with proper controls to bolster their conclusions.

Response: Thank you very much for your great suggestions. We appreciate your detailed feedback and apologize for not fully addressing the potential influence of endogenous HDAC6 protein in our initial manuscript. We demonstrate that the reduction in centrosomal HDAC6 localization during mitosis is crucial for maintaining normal ENKD1 centrosomal localization and proper spindle behavior. To further clarify this point, we quantified the amount of GFP-HDAC6 relative to endogenous HDAC6 in HCE-2 cells. Our results showed that the GFP-HDAC6 level was significantly higher than that of endogenous HDAC6 (Fig. EV3A,B). This elevated expression of exogenous HDAC6 likely leads to a more pronounced effect on the deacetylation of ENKD1, resulting in a significant reduction in its centrosomal localization and subsequent abnormalities in spindle behavior.

To address the complexity of interpreting experiments where wild-type or mutant HDAC6 is expressed in the presence of endogenous protein, we have conducted additional experiments with proper controls. Specifically, we expressed wild-type or mutant HDAC6 under conditions of endogenous HDAC6 knockdown to assess spindle behavior (Fig. 2N-Q) and ENKD1 localization (Fig. 4I,J). These experiments support our conclusion that overexpression of HDAC6 reduces ENKD1 acetylation and its localization at the centrosome, leading to abnormal spindle behavior.

Minor comments

1. Why does the nuclear size appear larger in cells overexpressing GFP-HDAC6 (Fig. 1E)?

Response: Thank you very much for your meticulous observation. We acknowledge the discrepancy in nuclear size observed in cells overexpressing GFP-HDAC6. To address this concern, we have carefully re-examined the data and provided more representative images in the revised manuscript (Fig. 1I).

2. To analyze the impact of HDAC6 OE on spindle orientation, authors should also perform live-imaging analysis to follow the behavior of the mitotic spindle. Similarly, it would be informative to know the spindle behaviors where ENKD1 wild-type and mutant version affect spindle angle (Fig. 5).

Response: We are deeply thankful for your valuable feedback. Following your suggestions, we have performed live-imaging analysis to follow the behavior of the mitotic spindle. Time-lapse microscopy demonstrated that the spindle angles in

HDAC6-overexpressed cells exhibited pronounced changes prior to anaphase onset compared to the controls (Fig. 2A-D). Similarly, overexpression of wild-type ENKDI or the K98Q mutant could rescue spindle orientation defects caused by ENKDI siRNAs, whereas the K98R mutant failed to rescue these defects (Fig. 5C,D and Appendix Fig. S4D,E).

Reviewer #2 (Remarks to the Author):

Manuscript by Song et al. compared the transcriptomic profiles of healthy and pathological corneas, and discovered that HDAC6 was dramatically upregulated in corneal diseases. They next showed that overexpression of HDAC6 in mice led to thickening of the corneal epithelium by impairing the spindle orientation and positioning in corneal epithelial cells. Further study validated that HDAC6 deacetylated ENKDI at K98 to compromise its interaction with γ -tubulin, thus impacting the centrosomal localization of ENKDI and its regulation in spindle dynamics. Overall, the study is well conducted with several interesting findings. Before considering this manuscript for publication, there are some points that the authors should address.

Response: We sincerely appreciate your constructive feedback and the recognition of the significance of our research. We acknowledge the need for revisions and have addressed the specific concerns you've raised to improve the manuscript. Thank you once again for your expert review.

1. One major concern is that previous studies have reported HDAC6 could directly regulate microtubule stability via deacetylation of alpha-tubulin. Thus, authors are suggested to clarify whether the affected spindle behavior in HDAC6-overexpressed corneal epithelial cells is involved in the deacetylation of microtubules by HDAC6.

Response: We thank the reviewer for this valuable suggestion. We acknowledge that previous studies have reported HDAC6 can directly regulate microtubule stability via deacetylation of α -tubulin. To address this concern, we transfected HCE-2 cells with HA-HDAC6 and GFP- α -tubulin wild-type or mutants, including K40Q (mutation of lysine 40 to glutamine to mimic acetylation) and K40R (mutation of lysine 40 to arginine to disrupt acetylation). Our results showed that there were no significant changes in the spindle behavior of HCE-2 cells transfected with GFP- α -tubulin wild type, K40Q, and K40R compared to those transfected with GFP (Appendix Fig. S2D-I). Thus, the deacetylation of microtubules by HDAC6 is unlikely involved in the affected spindle behavior in HDAC6-overexpressing corneal epithelial cells.

2. Fluorescence images of HDAC6-GFP are encouraged to provide for verification of HDAC6 localization at spindle poles in corneal epithelial cells.

Response: We thank the reviewer for highlighting the necessity of providing fluorescence images of GFP-HDAC6 to verify its localization at spindle poles in corneal epithelial cells. In the revised manuscript, we have included fluorescence

images of GFP-HDAC6 (Appendix Fig. S3A,B), which confirm its localization at centrosomes during interphase and spindle poles during metaphase. Additionally, we have provided the GFP channel in the fluorescence images involving HDAC6 overexpression (Fig. 2E,H,J,L,N,Q and Fig. 5E,K). These findings corroborate the HDAC6 localization at spindle poles in corneal epithelial cells.

3. Authors are recommended to further clarify in the Discussion why weakened interaction of ENKD1 with γ -tubulin would result in the defective spindle orientation and positioning.

Response: We deeply value the reviewer's constructive comment. Thank you for directing our attention to this significant aspect. We missed this important point in the initial discussion. To address this, we have amended the manuscript to include the following statement: The proper orientation and positioning of the mitotic spindle are critical for the accurate segregation of the genetic material and the maintenance of tissue architecture (Lechler & Mapelli, 2021). The centrosome, as the major microtubule-organizing center, plays a pivotal role in spindle assembly and orientation (Hoffmann, 2021). Our results indicate that the acetylation of ENKD1, modulated by HDAC6, is essential for maintaining its interaction with γ -tubulin. The ENKD1/ γ -tubulin interaction is crucial for the ability of ENKD1 to anchor at the centrosome and exert its function in spindle orientation and positioning, which in turn ensures proper cell division orientation and corneal homeostasis. However, the specific molecular mechanisms by which ENKD1 functions at the centrosome to regulate spindle behavior remain to be fully elucidated. ENKD1 may act as an important regulator to facilitate the nucleation and organization of microtubules, ensuring the correct alignment of the spindle. To gain a deeper understanding of these mechanisms, it will be necessary to analyze whether the microtubule-regulating capacity of ENKD1 is dependent on its centrosomal localization. Additionally, it will be important to dissect the precise molecular interactions involved in the above processes, including the binding affinity and structural changes of ENKD1 upon deacetylation.

4. Some English grammar and wording errors need to be revised.

Response: We apologize for the incorrect usage of English grammar and wording in our manuscript. In the revised manuscript, we have carefully addressed and corrected the errors. In terms of spelling, we specifically changed the "molelcular" to "molecular" in line 23, "mechnanistically" to "mechanistically" in line 77, "transfered" to "transferred" in line 592. In order to standardize the wording, we changed "outmost" to "outermost" in line 39. To make the sentence more appropriate, we changed "Meanwhile, it is a clear structure that can physically allow the passage of light into the eye and further focuses the light onto the retina for vision" to "Additionally, as a transparent structure, the cornea facilitates the passage of light into the eye and helps to focus light on the retina" in lines 42-43, and changed the "has" to "have" in lines 64. We believe that these amendments will enhance the scientific rigor and readability of our manuscript. Thank you for drawing our attention to these important details.

Reviewer #3 (Remarks to the Author):

The manuscript by Song et al examines the role of the deacetylase HDAC6 in regulating mitotic spindle behavior and corneal epithelial homeostasis. In a nutshell, the authors show that HDAC6 deacetylates the protein ENKD1, which impairs its interaction with γ -tubulin and centrosomal localisation. This thus reveals a critical role for HDAC6 in the control of corneal epithelial homeostasis.

In general the experiments are well executed and convincing. However, a number of points should be addressed before I can recommend the paper for publication. Below are specific points to address.

Response: We thank the reviewer for the positive comments about our manuscript and for helpful suggestions aimed at improving our manuscript. The following are specific responses to each question. Thank you for your expert opinions.

1. Figure 1: Based on publicly available transcriptomics data the authors identify overexpression of HDAC6 in corneal diseases samples. This observation is the basis for their subsequent experiments. Could the authors obtain some samples from corneal diseases and show by immunoblot that HDAC6 is overexpressed in these samples? This would strengthen the paper.

Response: We deeply appreciate your constructive comment. To strengthen our paper and provide additional evidence for the overexpression of HDAC6 in corneal diseases, we obtained samples from a bacterial keratitis (BK) mouse model (*Pseudomonas aeruginosa* keratitis). BK is a significant cause of painful corneal infections globally, leading to visual impairment and blindness. Consistent with the GEO dataset analysis, our immunoblot analysis of infected corneal epithelium showed a high expression of HDAC6 compared to healthy mice (Fig. 1F,G). These results demonstrate HDAC6 overexpression in corneal diseases, particularly in BK, and indicate a potential role for this protein in regulating corneal epithelial homeostasis.

2. Figure 3D: This shows a silver staining of proteins immunoprecipitated by a GFP antibody. Why is the antibody not visible? Were the proteins eluted from beads with stably bound antibody.

Response: We apologize for the lack of clarity in our initial manuscript regarding the GFP beads. We used GFP nanoab agarose beads (NuoyiBio, GNA-25-500) to perform the silver staining experiment. These beads do not contain the heavy and light chains of regular antibodies, which is the reason why the antibody itself is not visible in the silver staining. In the revised manuscript, we have added more details to the Materials and Methods section to clarify this method.

3. Fig. 3F and G: The immunoprecipitation does not appear to work, as HDAC6 is not visibly enriched in the IP vs the lysate. Can the authors comment and explain in the text?

Response: We sincerely apologize for this oversight. The issue arises from the low sample loading amount of the IP samples, which resulted in HDAC6 not being visibly enriched compared to the lysate. We have repeated the experiment with adjusted “IP: HDAC6” to match “lysate: HDAC6” (Fig. 3F,G).

4. Fig. 4B: This panel is a bit perplexing. It is shown that GFP-ENKD1 co-IP very well with HDAC6; this is unusual, as typically it is not possible to immunoprecipitate the substrates of an enzyme (or only very weakly if at all). Furthermore, when tubacin is added -which blocks the catalytic pocket of HDAC6- it does not decrease the co-IP; also that is strange, as one would expect that then there is less co-IP. Can the authors comment on this?

Response: Thank you for bringing this issue to our attention regarding Fig. 4B. We deeply regret the mistake in our manuscript. The confusion arises from a labeling mistake during the figure preparation process. Specifically, the label “IP: HA” in Fig. 4B should be corrected to “IP: GFP.” This error led to the perplexing results you noted, and we sincerely apologize for any misunderstanding it may have caused. To address this issue, we have corrected the labeling in the revised manuscript and meticulously rechecked all figures to ensure they are accurately represented.

5. General comments: 1. quantification of the immunoblots (including mentioning how many times the experiments have been done) would be useful to appreciate the magnitude of the effects observed.

Response: Thank you for your valuable suggestions. Following your recommendation, we have quantitatively analyzed the immunoblots in Fig. 1G, Fig. 4H, Fig. EV2D and Fig. EV4A-D. Additionally, we have detailed the number of times each experiment was repeated in the figure legends.

2. The methods section is not detailed enough; as it is now it is not possible to reproduce the experiments.

Response: We sincerely apologize for the lack of detail in the methods section, which could have hindered the reproducibility of our experiments. In the revised manuscript, we have provided a more detailed description of the experimental procedures in the Materials and Methods section.

Dear Dr. Liu,

Thank you for submitting your revised manuscript. It has now been seen by all of the original referees.

As you can see, all referees find that the study is significantly improved during revision and recommend publication. However, I need you to address the points below before I can accept the manuscript.

- Please address the remaining concerns of referee #1 (textually) and provide a point-by-point response. Please contact me if you would like to discuss any of these points further.
- Regarding the source data, source data files need to be saved as one zipped file/folder per figure - e.g. all the Source data files for figure 1 need to be saved in a single folder and this needs to be ZIPped and then uploaded as "SD figure 1.zip" file. All source data for EV Figures and/or appendix figures should be ZIPped together.
- Please provide 3-5 keywords for your study. These will be visible in the html version of the paper and on PubMed and will help increase the discoverability of your work.
- The Data Availability section is reserved for the primary datasets generated in the study deposited to external data repositories. If your study does not include such datasets, please replace the text in this section with the following statement: This study includes no data deposited in external repositories.
- Please rename the 'Competing interests' section as 'Disclosure Statement & Competing Interests'.
- Please remove the 'Author Contributions' section from the manuscript text.
- We note that the Appendix file is entitled as "Supplementary Information", which needs to be updated as Appendix. Moreover, a Table of Contents need to be added to the file along with page numbers.
- Please remove the Reagents & Tools table from the manuscript text and upload it as a separate file.
- 'Materials and Methods' section should be renamed as 'Methods'.
- 'Ethical statement' should not be a separate section. Instead, the statement should be moved to the Methods section.
- 'Additional information' section should be removed from the manuscript.
- During our routine image analysis, we note a potential duplication between Figure 1K and Figure EV2H. Please clarify. Also, please provide source data for EV2H.
- We note that an existing dataset (GSE241715), which is a part of PMID: 38274739 was reanalyzed in the current submission, which needs to be cited in the form of data citation as follows:

In the reference list:

Hörnberg E, Ylitalo EB, Crnalic S, Antti H, Stattin P, Widmark A, Bergh A, Wikström P (2011) Gene Expression Omnibus GSE29650 (<https://www.ncbi.nlm.nih.gov/geo/query/acc.cgi?acc=GSE29650>). [DATASET]

Hörnberg E, Ylitalo EB, Crnalic S, Antti H, Stattin P, Widmark A, Bergh A, Wikström P (2011) Expression of androgen receptor splice variants in prostate cancer bone metastases is associated with castration-resistance and short survival. PLoS One 6: e19059

In the main text, these datasets should be cited with the prefix "Data ref:" to distinguish them from the reference to the original article that reported the dataset. Example:

"...were grouped based on the relative levels of AR-Vs expressed, mainly AR-V7 (Hörnberg et al, 2011; Data ref: Hörnberg et al, 2011)."

- Our production/data editors have asked you to clarify several points in the figure legends:
 - o Please note that the exact p values are not provided in the legends of figures 1G, J, L, R; 2F, I, K, M, O, P; 3C, M; 4H, J, L; 5B, F, J, M; 6G, J, L; EV1 C-E; EV3 O; EV4A-D; EV5 C; supplementary figures 1C, 2C, 3B,
 - o Please indicate the statistical test used for data analysis in the legends of figures EV1 A, B.
 - o Please note that information related to n is missing in the legends of figures EV1 A.
 - o Please note that the red and green arrow is not defined in the legend of figure EV3 A. This needs to be rectified.
 - o Please note that the white arrows are not defined in the legend of figures 2C, E, J, N; 5A, D, E; supplementary figures 2D. This needs to be rectified.
 - o Please note that the white border is not defined in the legend of figures 2H, L, Q; 5H, K; EV3 J, N; supplementary figures 1B, G; 2G. This needs to be rectified.
- Papers published in EMBO Reports include a 'synopsis' and 'bullet points' to further enhance discoverability. Both are displayed on the html version of the paper and are freely accessible to all readers. The synopsis includes a short standfirst summarizing the study in 1 or 2 sentences (max 35 words) that summarize the paper and are provided by the authors and streamlined by the handling editor. I would therefore ask you to include your synopsis blurb and 3-5 bullet points listing the key experimental findings.
- In addition, please provide an image for the synopsis. This image should provide a rapid overview of the question addressed in the study but still needs to be kept fairly modest since the image size cannot exceed 550 (width) x 300-600 (height) pixels.

Thank you again for giving us to consider your manuscript for EMBO Reports, I look forward to your minor revision.

Kind regards,

Deniz Senyilmaz Tiebe

--

Deniz Senyilmaz Tiebe, PhD
Senior Scientific Editor
EMBO Reports

Referee #1:

I appreciate the authors' efforts to include controls and improve the manuscript. However, in certain cases, the interpretation of the data and microscopy images still remains unclear.

-The authors state that corneal epithelial thickness is significantly increased in mice injected with HDAC6 compared to controls (lines 130-133). As I previously mentioned, it appears that the total number of cells also increases in the HDAC6 overexpression condition (previous major point 3). Have the authors examined the total number of cells in these conditions? If cell cycle parameters are unchanged as shown by the authors, how do the authors explain the observed increase in corneal epithelial thickness?

-Similarly, I had earlier raised the possibility that the defects attributed to spindle orientation may, in fact, be due to cell misorientation, particularly if an increase in cell number alters tissue geometry and impacts cell positioning. In Fig. 1P, the authors depict a cell in anaphase within tissue expressing GFP-HDAC6. Does the spindle appear tilted due to the cell's positioning within the tissue, or is it a result of spindle misorientation? To substantiate their claim, the authors should analyze the spindle/DNA alignment during metaphase in such tissues. Would this align with the schematic shown in Fig. 1Q, or could it be that cell misorientation is being mistaken for spindle misorientation? Given that the image in Fig. 1P does not convincingly show spindle misorientation, I am not persuaded by the quantification provided in Fig. 1R. This is an important point as based on this authors decided to analyze the impact of HDAC6 OE on spindle misorientation.

-In Fig. 2C, the authors performed live-cell imaging in cells expressing GFP-Histone and SiR-tubulin to visualize spindle orientation. Interestingly, the control cell exhibits genomic instability, with some chromosomes located near the spindle pole. It is surprising to see the authors label 0 min as the onset of anaphase. How 0 min can be anaphase onset when imaged cells exhibit chromosomal instability? The spindle positioning assay in control and experimental samples should only be performed in cells with fully congressed chromosomes in metaphase; otherwise, the interpretation of the data may be compromised. What was the status of chromosome congression in the cells used for all the quantitative analyses shown in Fig. 2?

-The authors conclude that HDAC6 prominently localizes to the centrosome in HCE-2 cells during interphase, and that its localization diminishes during metaphase (Figs. 3B and 3C). Both in interphase and mitosis, HDAC6 also localizes in a punctate pattern, with these puncta being more prominent in cells overexpressing GFP-HDAC6 (Appendix Fig. S3). Is the localization of these puncta near the centrosome/gamma-tubulin coincidental? In Fig. 3B (lower pole), endogenous HDAC6 puncta do not consistently localize near the centrosome, with only one centrosome showing colocalization. Could the authors clarify this observation?

Referee #2:

Authors have adequately addressed my concerns, and I have no further comments.

Referee #3:

The issues I had raised have been taken care of carefully. In addition, my sense is that the issues of the other reviewers have also been addressed adequately.

I have no additional requests and I would recommend publication.

Point-by-Point Responses

Regarding the source data, source data files need to be saved as one zipped file/folder per figure - e.g. all the Source data files for figure 1 need to be saved in a single folder and this needs to be ZIPped and then uploaded as "SD figure 1.zip" file. All source data for EV Figures and/or appendix figures should be ZIPped together.

Response: We have made the changes as requested. Thanks.

Please provide 3-5 keywords for your study. These will be visible in the html version of the paper and on PubMed and will help increase the discoverability of your work.

Response: We have now supplied the keywords as instructed. Thanks.

The Data Availability section is reserved for the primary datasets generated in the study deposited to external data repositories. If your study does not include such datasets, please replace the text in this section with the following statement: This study includes no data deposited in external repositories.

Response: The statement has been updated as instructed. Thanks.

Please rename the 'Competing interests' section as 'Disclosure Statement & Competing Interests'.

Response: The section has been updated as instructed. Thanks.

Please remove the 'Author Contributions' section from the manuscript text.

Response: As instructed, the Author Contributions has been removed from the manuscript text. Thanks.

We note that the Appendix file is entitled as "Supplementary Information", which needs to be updated as Appendix. Moreover, a Table of Contents need to be added to the file along with page numbers.

Response: The file has been renamed as instructed. We have provided the Table of Contents in the appendix. Thanks.

Please remove the Reagents & Tools table from the manuscript text and upload it as a separate file.

Response: As instructed, the Reagents & Tools table has been removed from the manuscript text and upload it as a separate file. Thanks.

'Materials and Methods' section should be renamed as 'Methods'.

Response: The section has been renamed as 'Methods'. Thanks.

'Ethical statement' should not be a separate section. Instead, the statement should be moved to the Methods section.

Response: The Ethical statement has been moved to the Methods section. Thanks.

'Additional information' section should be removed from the manuscript.

Response: As instructed, the 'Additional information' section has been removed from the manuscript. Thanks.

During our routine image analysis, we note a potential duplication between Figure 1K and Figure EV2H. Please clarify. Also, please provide source data for EV2H.

Response: Thank you very much for bringing this to our attention. Upon reviewing the original data for Figure 1K and Figure EV2H, we identified that the duplication resulted from a copying error during figure preparation using Photoshop software. We sincerely apologize for this oversight. The image in Figure EV2H has been replaced with the correct one, and we have provided the uncropped images in the source data. Furthermore, we have meticulously rechecked all the source data for all experiments to ensure the reliability and integrity of the data.

We note that an existing dataset (GSE241715), which is a part of PMID: 38274739 was reanalyzed in the current submission, which needs to be cited in the form of data citation as follows.

Response: Thank you for the kind reminder. We have updated the References.

Please note that the exact p values are not provided in the legends of figures 1G, J, L, R; 2F, I, K, M, O, P; 3C, M; 4H, J, L; 5B, F, J, M; 6G, J, L; EV1 C-E; EV3 O; EV4A-D; EV5 C; supplementary figures 1C, 2C, 3B.

Response: We have provided the exact p values in the legends of these above figures. Thanks.

Please indicate the statistical test used for data analysis in the legends of figures EV1 A, B.

Response: Thank you for pointing this out. We have provided the statistical test used for data analysis in the legends of figures EV1 A, B.

Please note that information related to n is missing in the legends of figures EV1 A.

Response: Thank you for the kind reminder. We have provided the information related to n in the legend of figure EV1 A.

Please note that the red and green arrow is not defined in the legend of figure EV3 A. This needs to be rectified.

Response: We have defined the red and green arrow in the legend of figure EV3 A. Thanks.

Please note that the white arrows are not defined in the legend of figures 2C, E, J, N; 5A, D, E; supplementary figures 2D. This needs to be rectified.

Response: Thank you for bringing this to our attention. We have now defined the white arrows in the legends of Figures 2C, E, J, N; 5A, D, E; and Supplementary Figure 2D.

Please note that the white border is not defined in the legend of figures 2H, L, Q; 5H, K; EV3 J, N; supplementary figures 1B, G; 2G. This needs to be rectified.

Response: Thank you for pointing this out. We have now defined the white borders in the legends of Figures 2H, L, Q; 5H, K; EV3 J, N; and Supplementary Figures 1B, G, and 2G.

Papers published in EMBO Reports include a 'synopsis' and 'bullet points' to further enhance discoverability. Both are displayed on the html version of the paper and are freely accessible to all readers. The synopsis includes a short standfirst summarizing the study in 1 or 2 sentences (max 35 words) that summarize the paper and are provided by the authors and streamlined by the handling editor. I would therefore ask you to include your synopsis blurb and 3-5 bullet points listing the key experimental findings.

In addition, please provide an image for the synopsis. This image should provide a rapid overview of the question addressed in the study but still needs to be kept fairly modest since the image size cannot exceed 550 (width) x 300-600 (height) pixels.

Response: Requested information are as follows:

A) Short Summary

HDAC6 is upregulated in corneal diseases and causes ENKD1 deacetylation at lysine 98. Deacetylation of ENKD1 blocks its interaction with γ -tubulin and impairs its role in maintaining proper spindle behavior, thereby disrupting corneal epithelial homeostasis.

B) Key Results

1. HDAC6 is consistently upregulated in human corneal diseases and mouse bacterial keratitis models, correlating with epithelial thickening.
2. HDAC6 overexpression disrupts mitotic spindle orientation and positioning in corneal epithelial cells.
3. HDAC6 deacetylates ENKD1 at lysine 98, blocking its interaction with γ -tubulin and impairing its centrosomal role in maintain proper spindle behavior.

C) Synopsis image

Figure not shown.

Referee #1:

I appreciate the authors' efforts to include controls and improve the manuscript. However, in certain cases, the interpretation of the data and microscopy images still remains unclear.

-The authors state that corneal epithelial thickness is significantly increased in mice injected with HDAC6 compared to controls (lines 130-133). As I previously mentioned, it appears that the total number of cells also increases in the HDAC6 overexpression condition (previous major point 3). Have the authors examined the total number of cells in these conditions? If cell cycle parameters are unchanged as shown by the authors, how do the authors explain the observed increase in corneal epithelial thickness?

Response: Thank you for your thoughtful comment and for highlighting this important point. We sincerely appreciate the opportunity to clarify our findings.

We carefully examined the total number of corneal epithelial cells in mice injected with GFP or GFP-HDAC6 adenoviruses. Our analysis revealed that the total number

of corneal epithelial cells in HDAC6-overexpressing mice remained unchanged compared to control mice (representative images and quantitative data provided below).

These results align with our previous data showing that HDAC6 overexpression does not significantly affect cell proliferation or cell cycle progression. Instead, the observed increase in corneal epithelial thickness appears to result from the misorientation of cell division in the epithelium. Under normal conditions, corneal epithelial cells undergo symmetrical division, oriented either parallel or perpendicular to the basement membrane. However, HDAC6 overexpression disrupts this orientation, leading to the thickening of the corneal epithelium.

This mechanism is consistent with the schematic representation provided in Figure 1Q of our manuscript, which illustrates how HDAC6-mediated regulation of cellular division orientation contributes to epithelial homeostasis.

We hope this clarifies the underlying mechanism and addresses your concern. Thank you again for your valuable feedback.

Figure for referees not shown.

-Similarly, I had earlier raised the possibility that the defects attributed to spindle orientation may, in fact, be due to cell misorientation, particularly if an increase in cell number alters tissue geometry and impacts cell positioning. In Fig. 1P, the authors depict a cell in anaphase within tissue expressing GFP-HDAC6. Does the spindle appear tilted due to the cell's positioning within the tissue, or is it a result of spindle misorientation? To substantiate their claim, the authors should analyze the spindle/DNA alignment during metaphase in such tissues. Would this align with the schematic shown in Fig. 1Q, or could it be that cell misorientation is being mistaken for spindle misorientation? Given that the image in Fig. 1P does not convincingly show spindle misorientation, I am not persuaded by the quantification provided in Fig. 1R. This is an important point as based on this authors decided to analyze the impact of HDAC6 OE on spindle misorientation.

Response: Thank you very much for your meticulous observations and thoughtful review. We deeply appreciate your attention to detail and the opportunity to clarify our findings further.

Regarding your suggestion, we would like to address the possibility of cell misorientation due to changes in tissue geometry. As shown in the data provided

above, we found that HDAC6 overexpression does not significantly affect the total number of corneal epithelial cells. This suggests that changes in cell geometry due to cell number are not the primary cause of the observed effects.

In Figures 1P-R, our intention was to demonstrate that HDAC6 overexpression leads to changes in the orientation of cell division, rather than directly altering spindle orientation. Thus, the quantification in Figure 1R reflects the proportion of misoriented cells, not misoriented spindles.

The direction of cell division is determined by the orientation of the spindle (Taryn E Gillies and Clemens Cabernard, *Curr Biol.* 2011). To further investigate this, we analyzed the orientation of spindles in corneal cells during metaphase (Figures 2E-F). Our results confirm that HDAC6 overexpression indeed causes spindle misorientation in corneal epithelial cells, which aligns with the observed changes in cell division direction.

-In Fig. 2C, the authors performed live-cell imaging in cells expressing GFP-Histone and SiR-tubulin to visualize spindle orientation. Interestingly, the control cell exhibits genomic instability, with some chromosomes located near the spindle pole. It is surprising to see the authors label 0 min as the onset of anaphase. How 0 min can be anaphase onset when imaged cells exhibit chromosomal instability? The spindle positioning assay in control and experimental samples should only be performed in cells with fully congressed chromosomes in metaphase; otherwise, the interpretation of the data may be compromised. What was the status of chromosome congression in the cells used for all the quantitative analyses shown in Fig. 2?

Response: We sincerely apologize for the mistake. Thank you for noting that control cells exhibited genomic instability, with some chromosomes located near the spindle poles. We profoundly recognize that the determination of spindle positioning in control and experimental samples can only be conducted in cells with fully congressed chromosomes in metaphase. Therefore, we replaced the control group images with fully congressed chromosomes.

Additionally, we carefully examined the status of chromosome congression in the cells used for all the quantitative analyses shown in Fig. 2, and they were all in a fully condensed state.

-The authors conclude that HDAC6 prominently localizes to the centrosome in HCE-2 cells during interphase, and that its localization diminishes during metaphase (Figs. 3B and 3C). Both in interphase and mitosis, HDAC6 also localizes in a punctate pattern, with these puncta being more prominent in cells overexpressing GFP-HDAC6 (Appendix Fig. S3). Is the localization of these puncta near the centrosome/gamma-tubulin coincidental? In Fig. 3B (lower pole), endogenous HDAC6 puncta do not consistently localize near the centrosome, with only one centrosome showing colocalization. Could the authors clarify this observation?

Response: Thank you very much for your thoughtful question and for highlighting

this important aspect of our study. We co-stained HDAC6 with the centrosome marker γ -tubulin and observed co-localization. Although HDAC6 is distributed in a punctate manner throughout the cell, its localization at the centrosome is particularly prominent, consistent with findings reported in other studies (Pugacheva EN et al., *Cell*, 2007; Ainhoa Sánchez de Diego et al., *Nat Commun*, 2014.).

In this work, we further observed that HDAC6 exhibits dynamic changes in its centrosomal localization during the cell cycle. Specifically, HDAC6 shows strong centrosomal localization during interphase, while this localization becomes significantly weaker during mitosis. This dynamic behavior could explain the faint centrosomal staining of HDAC6 observed at the opposite spindle pole during metaphase (Fig. 3B, lower pole).

As noted by the reviewer, HDAC6 does not fully colocalize with γ -tubulin in our images. Specifically, the HDAC6 signal appears as a subset of the γ -tubulin staining, suggesting that HDAC6 localizes to a region within the centrosome that is smaller than the area covered by γ -tubulin. To precisely resolve the precise centrosomal localization of HDAC6, we plan to employ super-resolution microscopy techniques in future experiments.

We have added the following statement to the discussion section of our manuscript: “Furthermore, the precise centrosomal localization of HDAC6, as well as its spatiotemporal dynamics during cell cycle progression, remain to be fully elucidated. Unraveling these details will provide critical insights into the molecular network at the centrosome and its regulatory mechanisms underlying the spindle behavior.”

Min Liu
Nankai University
College of Life Sciences
94 Weijin Rd
Tianjin 300071
China

Dear Min,

Thank you for submitting your revised manuscript. I have now looked at everything and all is fine. Therefore, I am very pleased to accept your manuscript for publication in EMBO Reports.

Congratulations on a nice work!

Kind regards,

Deniz
--
Deniz Senyilmaz Tiebe, PhD
Senior Scientific Editor
EMBO Reports
